# Kernel Banzhaf: A Fast and Robust Estimator for Banzhaf Values

## Abstract

Banzhaf values provide a popular, interpretable alternative to the widely-used Shapley values for applications in explainable AI. Like Shapley values, computing Banzhaf values exactly requires time exponential in the number of inputs, necessitating the use of efficient estimators. In this work, we introduce Kernel Banzhaf, a regression-based estimator for Banzhaf values. Our approach leverages an existing regression formulation, whose exact solution corresponds to the exact Banzhaf values. Inspired by the success of Kernel SHAP for Shapley values, Kernel Banzhaf obtains an efficient approximation to the Banzhaf values by solving a subsampled instance of this regression problem. Across eight datasets, Kernel Banzhaf achieves substantially lower estimation error than prior methods and is more robust to noise, particularly at low-to-moderate noise levels or for larger numbers of features. We complement our experimental evaluation with strong theoretical guarantees on Kernel Banzhaf's performance.

## 1 Introduction

The increasing complexity of AI models has intensified challenges associated with model interpretability. Modern models, such as deep neural networks and ensemble methods, often make opaque decisions whose justification is not readily apparent to the user. This opacity makes it difficult to trust or rely on model predictions, especially in decision-making scenarios like healthcare, finance, and legal applications, which require rigorous justifications. Thus, reliable explainability tools are needed to bridge the gap between complex model behaviors and human understanding.

Among the various methods employed within explainable AI, game-theoretic approaches have gained prominence for quantifying the contribution of data features, both to the overall performance of a machine learning model, and to individual predictions made by the model. The most well-known game-theoretic approach is based on *Shapley values*, which provide a principled way to attribute the contribution of $n$ individual players to the outcome of a cooperative game, which is defined by a set function $v$ that maps every subset of $\{1, \ldots, n\}$ to a real value (Shapley, 1953).

In the context of feature attribution, each "player" is a feature and $v$ maps a subset of features to, e.g., the overall test loss when the chosen features are used, or the prediction made for a specific individual given access to the chosen features. The Shapley value quantifies the average marginal contribution of a feature on the set function, computed as the weighted average over all possible combinations of features included in the model (Lundberg & Lee, 2017). More formally, the Shapley value $\phi_i$ of a player $i \in \{1, \ldots, n\}$ is

$$\phi_i^{\text{shap}} = \frac{1}{n} \sum_{S \subseteq [n] \setminus \{i\}} \binom{n-1}{|S|}^{-1} [v(S \cup \{i\}) - v(S)]. \tag{1}$$

While primarily associated with feature attribution (Lundberg & Lee, 2017; Karczmarz et al., 2022), game-theoretic methods like Shapley values are also used in machine learning tasks like feature selection (Covert et al., 2020) and data valuation (Ghorbani & Zou, 2019; Wang & Jia, 2023).

An alternative to Shapley values is the Banzhaf value, which also measures each individual's contribution to an overall outcome (Banzhaf, 1965; Lehrer, 1988). While Shapley values are more widely used, Banzhaf values are often considered more intuitive for AI applications because they treat each subset of players as equally important, instead of weighting by a function of the subset size, as in Equation (1) for Shapley values. In particular, the Banzhaf value is

$$\phi_i^{\text{banz}} = \frac{1}{2^{n-1}} \sum_{S \subseteq [n] \setminus \{i\}} [v(S \cup \{i\}) - v(S)]. \tag{2}$$

It has also been observed that Banzhaf values tend to be more robust than Shapley values in the context of explainable AI (Karczmarz et al., 2022; Wang & Jia, 2023). These benefits have led to the use of Banzhaf values for feature attribution (Datta et al., 2015; Kulynych & Troncoso, 2017; Sliwinski et al., 2018; Patel et al., 2021) and, more recently, for data valuation Wang & Jia (2023); Li & Yu (2024c).

For general set functions, the exact computation of both Shapley and Banzhaf values requires at least $2^n$ time, where $n$ is the number of features in our problem. In particular, we must evaluate $v(S)$ for every possible subset $S \subseteq \{1, \ldots, n\}$. Accordingly, the problem of *approximating* Shapley values at lower cost has been widely studied. A leading estimator for Shapley values is KernelSHAP (Lundberg & Lee, 2017), a model-agnostic technique that leverages a connection to linear regression, approximating Shapley values by solving a sampled weighted least squares problem (Charnes et al., 1988; Lundberg & Lee, 2017). KernelSHAP has been further improved with paired sampling (Covert & Lee, 2021) and leverage score sampling (Musco & Witter, 2025).

In contrast to Shapley values, fewer algorithms have been proposed to approximate Banzhaf values for arbitrary set functions. For tree-structured set functions, such as those corresponding to decision tree based models, exact Banzhaf values can be efficiently computed (Karczmarz et al., 2022). For general set functions, such as those defined by neural-network models, Monte Carlo sampling (i.e., estimating the sum in Equation (2) from a subsample) can be used to estimate each Banzhaf value separately (Merrill III, 1982; Bachrach et al., 2010). The Maximum Sample Reuse (MSR) algorithm is a variant of Monte Carlo sampling that reuses samples for the estimates of different Banzhaf values (Wang & Jia, 2023). However, even with reuse, the naive Monte Carlo estimator typically requires a large number of samples (which equates to a large number of evaluations of $v$) to obtain meaningfully accurate estimates for the Banzhaf values.

**Our Contributions.** In this work, we show that, like Shapley values, Banzhaf values can be approximated far more accurately through *approximate regression*. To do so, we leverage an existing regression formulation (concretely, a linear least squares regression problem involving $n$ variables) whose solution exactly corresponds to the Banzhaf values (Hammer & Holzman, 1992). We show that this problem can be efficiently solved by subsampling rows of the regression problem, each of which corresponds to a subset of players and requires just a single evaluation of the set function $v$.

This yields a new estimator, *Kernel Banzhaf*, that substantially improves on existing Monte Carlo methods. In particular, we conduct an extensive experimental evaluation of Kernel Banzhaf in Section 4, assessing performance in terms of sample efficiency and robustness to noise in the evaluations of $v$. Furthermore, we evaluate Kernel Banzhaf in a downstream feature-ranking task. Our experiments span tree-based models and neural networks. For a fixed number of evaluations of $v$, Kernel Banzhaf typically reduces Banzhaf-value estimation error by an order of magnitude or more and is more robust at low-to-moderate noise levels or for larger numbers of features (see, e.g., Figure 1).

We complement our experimental evaluation of Kernel Banzhaf with strong theoretical guarantees. We prove that, with just $O(n \log n)$ evaluations of $v$, Kernel Banzhaf returns a solution with bounded squared $\ell_2$-norm difference to the exact Banzhaf values (see Theorem 3.3). Obtaining similar theoretical guarantees for Monte Carlo sampling and improvements like Maximum Sample Reuse requires making a strong assumption on the maximum magnitude of $v(S)$. See Section 3.3 for further comparison of theoretical results.

Our main contributions can be summarized as follows:

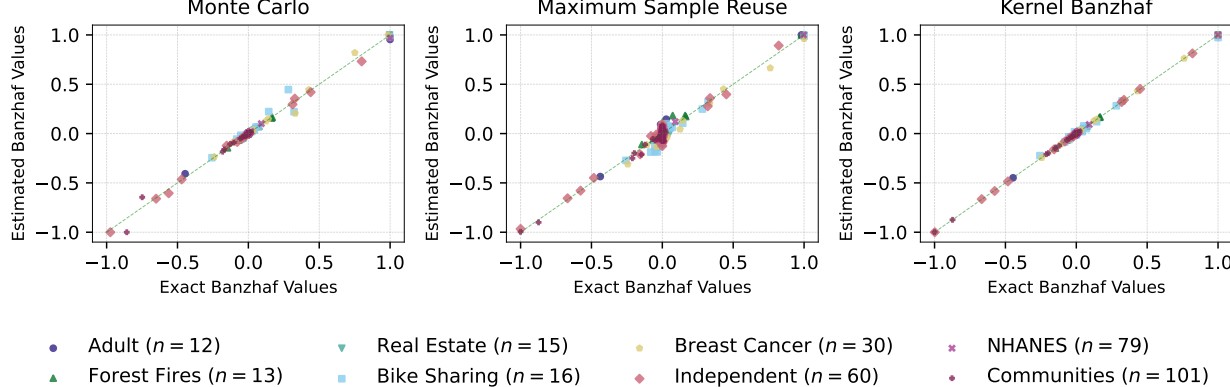

Figure 1: Comparison of exact and estimated Banzhaf feature attribution values across eight datasets, where every estimator uses $m = 20n$ evaluations of the set function $v$. The evaluations dominate the computational cost of each algorithm. Each plot, labeled with its estimator, shows normalized estimated versus exact Banzhaf values across all features for a randomly selected data point from each dataset. Points closer to the diagonal line indicate more accurate estimates; the plots suggest that Kernel Banzhaf is more accurate than the Monte Carlo and Maximum Sample Reuse estimators.

1. We propose Kernel Banzhaf, a regression-based approximation algorithm for estimating Banzhaf values for general set functions. The underlying regression equivalence (Theorem 3.1) is classical: Hammer & Holzman (1992) state it for simple games (their Corollary 3.1), and the general-set-function version follows from their approximation theory (see also Grabisch et al., 2000; Marichal & Mathonet, 2011). We include a short self-contained proof in a simplified parameterization; our contribution is the estimator built on it and its analysis.

2. We show that Kernel Banzhaf provably returns accurate solutions. Our theoretical guarantees depend on the magnitude of the true Banzhaf values rather than the magnitude of the set function, as for prior estimators.

3. Across eight datasets and several evaluation settings, Kernel Banzhaf achieves substantially lower estimation error than prior methods.

## 2 Background

Let $n$ be a positive integer and define $[n] := \{1, 2, \ldots, n\}$. We let $2^{[n]}$ denote the set of all $2^n$ possible subsets of $[n]$ (including the emptyset). For set $S$, define the indicator vector $\mathbb{1}[S] \in \{0, 1\}^n$ so that $\mathbb{1}[S]_i = 1$ if $i \in S$ and 0 otherwise. Let $v : 2^{[n]} \to \mathbb{R}$ be a set function. Shapley and Banzhaf values are as defined in Equations (1) and (2) in the prior section. For notational simplicity, we will use $\phi_i = \phi_i^{\text{banz}}$ in the rest of the paper. Let $\boldsymbol{\phi} \in \mathbb{R}^n$ denote the vector $[\phi_1, \ldots, \phi_n]$.

Since we make no structural assumptions about $v$, we consider estimators for computing Banzhaf values that access the function via *samples*; formally, a single sample corresponds to evaluating $v(S)$ for any $S$ the algorithm chooses.

**Banzhaf Values for Feature Attribution.** As discussed in the previous section, Shapley and Banzhaf values have a number of applications in machine learning, which correspond to different choices of the set function $v$. For example, we can consider feature selection by defining $v(S)$ as the loss of a model trained only on the features in $S$. Similarly, we can consider data selection by defining $v(S)$ as the loss of a model trained only on the observations in $S$.

One of the most popular applications of Banzhaf values in explainable AI is *feature attribution* (also known as feature importance and feature influence). Let $\mathcal{M} : \mathbb{R}^n \to \mathbb{R}$ be a trained model which takes input $\mathbf{x} \in \mathbb{R}^n$. Our goal is to explain the predictions of the model on a particular data point $\mathbf{x}$. Given a subset

of features $S$, define $\mathbf{x}^S$ as the observation where $\mathbf{x}_i^S = \mathbf{x}_i$ if feature $i \in S$ and, otherwise, $\mathbf{x}_i^S$ is sampled from the data distribution (possibly conditioned on the features in $S$). Then $v(S) = \mathbb{E}[\mathcal{M}(\mathbf{x}^S)]$, where the expectation is over the sampling of the features $i \notin S$. This expectation can typically only be estimated via sampling, meaning that we must contend with noisy access to the value function $v$. This setting motivates our evaluation of the robustness of Banzhaf value estimation methods in Section 4.

## 3   Kernel Banzhaf

The starting point of our work is a formulation of Banzhaf values in terms of a structured linear regression problem with $n$ variables and an exponential number of rows (Hammer & Holzman, 1992). In Hammer & Holzman (1992), the authors show in Corollary 3.1 that the $n$ parameter solution to the full regression problem exactly corresponds to the $n$ Banzhaf values when the value function is simple (monotone and binary). For completeness, we show Theorem 3.1, which establishes the same result for *all* value functions; the general-case statement also follows from their Theorem 2.3, and appears in the later least-squares approximation literature (Grabisch et al., 2000; Marichal & Mathonet, 2011).

Of course, solving an exponentially large regression problem is infeasible; hence, we propose Kernel Banzhaf, which approximately solves the regression problem by subsampling just a small subset of rows. We establish theoretical guarantees on how well the resulting solution approximates the Banzhaf values. We compare the Kernel Banzhaf to other Banzhaf value estimators in Section 4.

### 3.1   Linear Regression Formulation

Let $\mathbf{A} \in \mathbb{R}^{2^n \times n}$ be a design matrix and $\mathbf{b} \in \mathbb{R}^{2^n}$ be a target vector. We will use subsets $S \subseteq [n]$ to index the rows of $\mathbf{A}$ and entries of $\mathbf{b}$. In particular, $[\mathbf{A}]_{S,i}$ denotes the $i^{\text{th}}$ entry of row $S$. We set[1]:

$$[\mathbf{A}]_{S,i} = \begin{cases} +\frac{1}{2} & \text{if } i \in S \\ -\frac{1}{2} & \text{if } i \notin S \end{cases} \quad \text{and } \mathbf{b}_S = v(S). \tag{3}$$

The following theorem states this connection between the regression problem and Banzhaf values precisely.

**Theorem 3.1** (Linear Regression Equivalence). *Consider $\mathbf{A}$ and $\mathbf{b}$ as defined in Equation 3. Let*

$$\mathbf{x}^* = \arg\min_{\mathbf{x}} \|\mathbf{A}\mathbf{x} - \mathbf{b}\|_2^2.$$

*Then $\boldsymbol{\phi} = \mathbf{x}^*$, where $\boldsymbol{\phi}$ are the Banzhaf values of $v$.*

*Proof of Theorem 3.1.* As is standard, we have that:

$$\arg\min_{\mathbf{x}} \|\mathbf{A}\mathbf{x} - \mathbf{b}\|_2^2 = (\mathbf{A}^\top \mathbf{A})^{-1} \mathbf{A}^\top \mathbf{b}.$$

We will analyze the right hand side. The $(i, j)$ entry in $\mathbf{A}^\top \mathbf{A}$ is given by

$$[\mathbf{A}^\top \mathbf{A}]_{i,j} = \sum_{S \subseteq [n]} \left( \mathbb{1}[i \in S] - \frac{1}{2} \right) \left( \mathbb{1}[j \in S] - \frac{1}{2} \right). \tag{4}$$

If $i \neq j$, there are $2^{n-1}$ terms of $-\frac{1}{4}$ when $\mathbb{1}[i \in S] \neq \mathbb{1}[j \in S]$ and $2^{n-1}$ terms of $+\frac{1}{4}$ when $\mathbb{1}[i \in S] = \mathbb{1}[j \in S]$, hence Equation 4 is 0. If $i = j$, we have $2^n$ terms of $+\frac{1}{4}$ hence Equation 4 is $2^{n-2}$. Together, this gives that

$$\mathbf{A}^\top \mathbf{A} = 2^{n-2} \mathbf{I}. \tag{5}$$

---

[1]Hammer & Holzman (1992) use a more complicated regression formulation. For ease of understanding, we simplify it slightly.

Then $(\mathbf{A}^\top \mathbf{A})^{-1} = \frac{1}{2^{n-2}}\mathbf{I}$. Continuing, we have

$$(\mathbf{A}^\top \mathbf{A})^{-1}\mathbf{A}^\top \mathbf{b} = \frac{1}{2^{n-2}}\mathbf{A}^\top \mathbf{b} = \frac{1}{2^{n-2}} \sum_{S \subseteq [n]} \left( \mathbb{1}[S] - \frac{1}{2}\mathbf{1} \right) v(S).$$

We can write the $i^{\text{th}}$ entry of the above as:

$$[(\mathbf{A}^\top \mathbf{A})^{-1}\mathbf{A}^\top \mathbf{b}]_i = \frac{1}{2^{n-2}} \sum_{S \subseteq [n]} \left( \mathbb{1}[i \in S] - \frac{1}{2} \right) v(S) = \frac{1}{2^{n-1}} \sum_{S \subseteq [n] \setminus \{i\}} v(S \cup \{i\}) - v(S),$$

which is exactly Equation 2. The statement follows. $\qquad\square$

**Another Regression Formulation.** Ruiz et al. (1998) introduce a regression problem with the property that $\phi = \mathbf{x}^* + c\mathbf{1}$ for $\mathbf{1} \in \mathbb{R}^n$ the all-ones vector and $c$ an unknown number that depends on $v$ through the sum of the unknown Banzhaf values, so $c$ cannot be read off the regression solution (Li & Yu, 2024b). Li & Yu (2024a) circumvent the shift by augmenting the game with a dummy player whose Banzhaf value is 0; we include the resulting GELS estimator as a baseline in Section 4. Marichal & Mathonet (2011) generalize this least-squares characterization in a complementary direction: replacing the uniform weighting over subsets with a product probability distribution, they obtain weighted Banzhaf power and interaction indexes as the leading coefficients of the best weighted low-degree approximation of $v$. Their uniform-weight, degree-one case coincides with the equivalence in Theorem 3.1; their study is representational and axiomatic, and does not address estimating the indexes from few evaluations of $v$, which is the problem we take up.

### 3.2 The Kernel Banzhaf Algorithm

Since $\mathbf{A}$ and $\mathbf{b}$ have $2^n$ rows, constructing the linear regression problem in Theorem 3.1 to calculate the Banzhaf values is computationally prohibitive. In particular, we must evaluate $v(S)$ for all $2^n$ subsets of $[n]$ to compute $\mathbf{b}$. Inspired by Kernel SHAP, we avoid this cost by constructing a much smaller regression problem that contains a subsample of $m \ll 2^n$ rows from $\mathbf{A}$ and $\mathbf{b}$. Let the subsampled design matrix be $\tilde{\mathbf{A}} \in \mathbb{R}^{m \times n}$ and the subsampled target vector be $\tilde{\mathbf{b}} \in \mathbb{R}^m$. The estimate we produce is

$$\hat{\phi} = \arg\min_{\mathbf{x}} \|\tilde{\mathbf{A}}\mathbf{x} - \tilde{\mathbf{b}}\|_2^2.$$

The most direct approach to constructing $\tilde{\mathbf{A}}$ and $\tilde{\mathbf{b}}$ would be to subsample $m$ rows from $\mathbf{A}$ and $\mathbf{b}$ uniformly at random. We do so with a slight modification – whenever the row corresponding to a set $S$ is sampled, we also select the row corresponding to $[n] \setminus S$. Referred to as "paired sampling", this approach has been used in the Kernel SHAP algorithm for Shapley values Covert & Lee (2021). While it has no impact on our theoretical results (unpaired sampling would give almost identical bounds) we observe an experimental improvement for Banzhaf value estimation as well. The efficacy of paired sampling has been widely observed, and recent work provides a compelling explanation Fumagalli et al. (2026a;b): every set function decomposes into even and odd components, and Fumagalli et al. (2026a) show that Shapley values depend only on the odd component, which paired sampling isolates by filtering out the irrelevant even component. It is easy to check that Banzhaf values also only depend on the odd component.[2] Detailed pseudocode for our Kernel Banzhaf appears in Algorithm 1.

**Why Solve a Regression Problem?** It is tempting to think that Algorithm 1 could be simplified by skipping the least-squares solve and instead directly rescaling the subsampled rows, i.e., returning $\frac{4}{m}\tilde{\mathbf{A}}^\top \tilde{\mathbf{b}}$. This shortcut is *not* generally equivalent to Algorithm 1 at finite $m$: the identity $(\mathbf{A}^\top \mathbf{A})^{-1}\mathbf{A}^\top \mathbf{b} = \frac{1}{2^{n-2}}\mathbf{A}^\top \mathbf{b}$ used in the proof of Theorem 3.1 relies on $\mathbf{A}^\top \mathbf{A} = 2^{n-2}\mathbf{I}$ holding exactly for the full population of $2^n$ rows. For a random subsample, $\tilde{\mathbf{A}}^\top \tilde{\mathbf{A}}$ is proportional to $\mathbf{I}$ in expectation, since $\mathbb{E}[\tilde{\mathbf{A}}^\top \tilde{\mathbf{A}}] = \frac{m}{4}\mathbf{I}$, but the realized matrix need not be exactly proportional. Solving $\arg\min_{\mathbf{x}} \|\tilde{\mathbf{A}}\mathbf{x} - \tilde{\mathbf{b}}\|_2^2$ accounts for this finite-sample discrepancy; directly rescaling $\tilde{\mathbf{A}}^\top \tilde{\mathbf{b}}$ does not. The same phenomenon is documented for Shapley

---

[2]An even function satisfies $f(S) = f(S^c)$ for all pairs consisting of $S$ and the complement $S^c = [n] \setminus S$. Then, for an even function, $\phi_i = 0$ since $f(S)$ and $f(S^c)$ have the same (uniform) weight but opposite sign.

---

**Algorithm 1** Kernel Banzhaf

---

1: **Input:** $n$: positive integer, $v : \{0,1\}^n \to \mathbb{R}$: set function, $m$ : even number of samples such that $n < m \leq 2^n$
2: **Output:** $\hat{\phi} \in \mathbb{R}^n$: estimated Banzhaf values
3: Initialize $\tilde{\mathbf{A}} \leftarrow \mathbf{0}_{m \times n}$
4: **for** $j \in \{1, 3, 5, \ldots, m-1\}$ **do**
5:     Sample $S_j$ uniformly from all subsets of $[n]$
6:     Set $\tilde{\mathbf{A}}_j \leftarrow \mathbb{1}[S_j] - \frac{1}{2}\mathbf{1}$
7:     Compute $S_{j+1} = [n] \setminus S_j$                                        // Paired sampling
8:     Set $\tilde{\mathbf{A}}_{j+1} \leftarrow \mathbb{1}[S_{j+1}] - \frac{1}{2}\mathbf{1}$
9: **end for**
10: Compute $\tilde{\mathbf{b}} \leftarrow [v(S_1), \ldots, v(S_m)]$
11: Solve $\hat{\phi} \leftarrow \arg\min_{\mathbf{x}} \|\tilde{\mathbf{A}}\mathbf{x} - \tilde{\mathbf{b}}\|_2$
12: **Return:** $\hat{\phi}$

---

values: Covert & Lee (2021) proposed the analogous plug-in estimator (unbiased KernelSHAP) and observed that it converges substantially more slowly than solving the subsampled regression, and Chen et al. (2025) prove sample complexity bounds that separate the two approaches: their bound for the regression estimator depends on $\mathbf{b}$ only through its residual off the column span of the design matrix, while their bound for the plug-in estimator depends on all of $\mathbf{b}$. Under complement pairing, the rescaled estimator is exactly paired-sampling Maximum Sample Reuse, which we include as *MSR (paired)* in Section 4. As shown in Figure 2 and other experiments, this algorithm performs substantially worse than Kernel Banzhaf. See Appendix H for more details.

**Time Complexity.** Let $T$ be the time complexity of evaluating the set function $v$. The total runtime of Kernel Banzhaf is $O(Tm + mn^2)$. Constructing $\tilde{\mathbf{A}}$ takes $O(mn)$ time, evaluating $v$ on the sampled subsets takes $O(Tm)$ time, and solving the $m \times n$ least-squares problem takes $O(mn^2)$ time. For most applications in ML and explainable AI, we expect the $O(Tm)$ term to dominate. For example, a forward pass on a fully connected neural network whose hidden width scales with the number of features $n$ takes $\Omega(n^2)$ time, in which case the $O(Tm)$ evaluation cost already matches the $O(mn^2)$ cost of the regression solve. Figure 6 in Appendix D confirms that the cost of Kernel Banzhaf (and prior Monte Carlo methods) scales roughly linearly with the number of samples, $m$, i.e., the number of evaluations of $v$. Kernel Banzhaf and MSR have comparable wall-clock time in our experiments (Appendix D).

### 3.3 Approximation Guarantees

Despite its relative simplicity, Kernel Banzhaf admits strong approximation guarantees. Related guarantees for regression-based Shapley estimation use results on subsampled regression from randomized numerical linear algebra Musco & Witter (2025); Woodruff et al. (2014); Drineas & Mahoney (2018). In Appendix A, we use similar tools to show that a near-linear number of samples suffices to solve the Banzhaf regression problem near-optimally:

**Theorem 3.2.** *If $m = O(n \log \frac{n}{\delta} + \frac{n}{\delta \epsilon})$, Algorithm 1 produces an estimate $\hat{\phi}$ that satisfies, with probability $1 - \delta$,*

$$\|\mathbf{A}\hat{\phi} - \mathbf{b}\|_2^2 \leq (1 + \epsilon)\|\mathbf{A}\phi - \mathbf{b}\|_2^2. \tag{6}$$

Chen & Price (2019) give a known lower bound on the sample complexity of agnostic active linear regression that shows Theorem 3.2 is near optimal, made precise below. For constant $\epsilon$ and $\delta$ this is intuitive: in particular, consider a linear set function where $\mathbf{b}$ is in the span of $\mathbf{A}$. Then the right-hand side is 0 and we must recover $\phi$ exactly. To do this, we need to observe at least $n$ linearly independent rows of the regression problem. More precisely, Chen & Price (2019) prove a lower bound (their Theorem 8.4; theorem numbering follows the arXiv version) specifically for algorithms restricted to i.i.d. sampling from a fixed row distribution, the class Algorithm 1 belongs to: any such algorithm needs $\Omega(K \log n + K/\epsilon)$ samples,

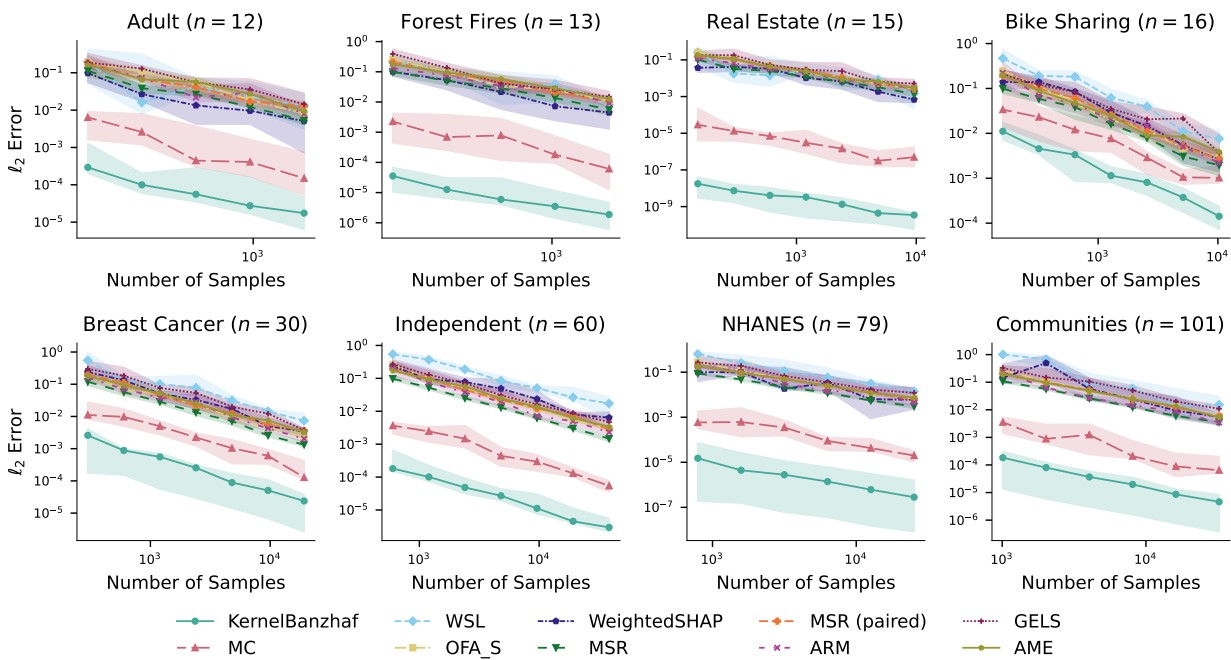

Figure 2: Relative squared $\ell_2$-norm error ($\|\hat{\boldsymbol{\phi}} - \boldsymbol{\phi}\|_2^2/\|\boldsymbol{\phi}\|_2^2$) across increasing sample sizes on eight datasets. Each point is the median over 10 runs, one per explicand; shaded regions show the 25th to 75th percentiles. Kernel Banzhaf outperforms the other estimators by several orders of magnitude.

where $K = \sup_x \sup_{h \in \mathcal{F}} |h(x)|^2/\|h\|_D^2$ is a condition number, and it follows from their Lemma 6.5 that $K \geq n$, with equality exactly when leverage scores are uniform across rows. Our matrix $\mathbf{A}$ achieves this minimum, since every row has the same leverage score $\ell_S = n/2^n$ (shown at the start of Appendix A), so Theorem 3.2's $O(n \log n + n/\epsilon)$ upper bound is minimax optimal among i.i.d.-sampling algorithms. We note that this optimality is only within that restricted class: Chen & Price's own algorithm removes the $\log n$ factor entirely (their Theorem 1.1, matched by a general lower bound in their Theorem 8.1) by actively selecting rows rather than sampling them i.i.d., so a variant of Kernel Banzhaf with actively chosen samples could in principle improve on Algorithm 1 by a $\log n$ factor. We leave exploring such actively sampled variants to future work.

As a direct consequence of Theorem 3.2, we obtain a bound on the Euclidean distances between the approximate Banzhaf values computed by our Kernel Banzhaf method and the true values. In particular, in Appendix A, we show:

**Corollary 3.3.** *Let $\gamma = \|\mathbf{A}\boldsymbol{\phi} - \mathbf{b}\|_2^2/\|\mathbf{A}\boldsymbol{\phi}\|_2^2$. Any $\hat{\boldsymbol{\phi}}$ that satisfies Equation 6 also satisfies*

$$\|\hat{\boldsymbol{\phi}} - \boldsymbol{\phi}\|_2^2 \leq \epsilon\gamma\|\boldsymbol{\phi}\|_2^2. \tag{7}$$

*The reverse is also true: Equation 7 implies Equation 6.*

Corollary 3.3 incurs a dependence on a parameter $\gamma$, which captures the extent to which $v$ deviates from an additive set function (geometrically, the distance of $\mathbf{b}$ from the linear span of the rows of $\mathbf{A}$). Because there is no relaxation in the guarantee from Theorem 3.2 to Corollary 3.3—and given that the sample complexity of Theorem 3.2 is optimal among i.i.d.-sampling algorithms (as made precise above)—we argue that dependence on $\gamma$ is inherent to the Kernel Banzhaf method.

This geometric picture has a natural pseudo-Boolean Fourier interpretation. The rows of $\mathbf{A}$ are, up to scaling, the degree-1 Fourier characters, and $\boldsymbol{\phi}$ is the projection of $\mathbf{b}$ onto their span. Thus $\|\mathbf{A}\boldsymbol{\phi}\|_2^2$ is the degree-1 Fourier mass, while $\|\mathbf{A}\boldsymbol{\phi} - \mathbf{b}\|_2^2$ is the mass in the constant term and interactions of degree at

least two. Hence $\gamma$ is exactly the ratio of Fourier mass outside degree 1 to degree-1 mass, connecting it to interaction-based analyses of Shapley-value estimation (Fumagalli et al., 2026a;b).

Experimentally, we find this quantity is manageable: computing $\gamma$ exactly by full enumeration, its median value was approximately 7 for the datasets small enough for us to compute it, with per-dataset medians between 2.7 and 18.2. While this magnitude implies that the sample complexity must scale with $\gamma$ (requiring $\epsilon < \frac{1}{7}$ and roughly $m = \frac{n}{\epsilon} = 7n$ samples for non-trivial bounds in this regime), this cost appears as a data-dependent constant factor. Consequently, the method remains computationally efficient and polynomial in $n$, avoiding the exponential scaling required for exact computation. Moreover, about 99% of the residual mass lies in the constant Fourier component at the median, which complement pairing cancels exactly; the odd-degree residual mass that drives finite-sample error under pairing has a median below 0.4% of the residual mass (Appendix H, Table 3). This decomposition helps explain why the observed errors are much smaller than a worst-case reading of Corollary 3.3. In the ideal case where $\mathbf{b}$ lies exactly in the span of $\mathbf{A}$, $\gamma$ vanishes ($\gamma = 0$) and we recover the exact Banzhaf values. We also stress-test the dependence on $\gamma$ directly, on synthetic set functions with a known Fourier spectrum whose interaction mass we dial from $\gamma = 0$ up to $\gamma = 1000$ (Appendix G).

**Comparison of Theoretical Guarantees.** Basic concentration bounds can be used to prove similar guarantees for the Monte Carlo and Maximum Sample Reuse estimators (see e.g., Theorems 4.8 and 4.9 in Wang & Jia (2023)). When MC is run with $O\left(\frac{n^2}{\epsilon}\log(\frac{n}{\delta})\right)$ samples and when MSR is run with $O\left(\frac{n}{\epsilon}\log(\frac{n}{\delta})\right)$ samples, they respectively return estimates $\tilde{\phi}$ that satisfy, with high probability,

$$\|\hat{\phi} - \phi\|_2^2 \leq \epsilon \max_{S \subseteq [n]} v(S)^2.$$

That is, both prior methods obtain a bound that depends on the maximum of $v$, rather than $\|\phi\|_2^2$.[3] In general, we expect the maximum magnitude of the set function to be larger than the Banzhaf values, which measure an *average* marginal contribution. This is consistent with the lower empirical error of Kernel Banzhaf (see, e.g., Table 1).

### 3.4 Extension to Probabilistic Values

Banzhaf and Shapley values can be generalized to so-called *probabilistic values*, which have also found applications in machine learning Kwon & Zou (2022a); Li & Yu (2024b;c). In Appendix B, we show how the regression formulation of Section 3.1 can be extended to probabilistic values, and prove an approximation guarantee (Theorem B.2) that recovers Corollary 3.3 as a special case. We also report an exploratory evaluation across nine weightings.

## 4 Experiments

We evaluate Banzhaf estimators across eight datasets[4]. While Banzhaf values can be applied in many settings by changing the value function $v$, we focus on the canonical task of feature attribution. In addition to its popularity, feature attribution allows us to compute exact Banzhaf values even for large $n \geq 30$ when the machine learning model is a tree (Karczmarz et al., 2022). For datasets with $n \leq 20$—where Banzhaf values can be exhaustively computed using Equation 2—we define the value function as the prediction made by a general neural network. We expand on the datasets and models in Appendix C.

With the exact Banzhaf values in hand, we compare the performance of Kernel Banzhaf to other Banzhaf value estimators. Our primary metric is relative squared error, $\|\hat{\phi} - \phi\|_2^2 / \|\phi\|_2^2$. In addition, we compare estimators in terms of how accurately they recover the *rank* of non-zero Banzhaf values. We measure the

---

[3] Wang & Jia (2023) state their theorems under the assumption $v : 2^{[n]} \to [0, 1]$, in which case the factor $\max_S v(S)^2$ is at most 1; the form stated here follows by normalizing $v$ by $\max_S |v(S)|$.

[4] We will make our code publicly accessible after publication.

Table 1: The relative squared $\ell_2$-norm error (i.e., $\|\hat{\phi} - \phi\|_2^2/\|\phi\|_2^2$) of Banzhaf estimators with $m = 40n$ samples over 10 runs. We bold the lowest error for the 25% percentile, median, and 75% percentile. Across all eight datasets and every summary statistic, Kernel Banzhaf returns estimates with the lowest error, typically by an order of magnitude.

| | Adult | Forest Fires | Real Estate | Bike Sharing | Breast Cancer | Independent | NHANES | Communities | Mean |
|---|---|---|---|---|---|---|---|---|---|
| **KernelBanzhaf** | | | | | | | | | |
| Mean | $\mathbf{1.50 \times 10^{-4}}$ | $\mathbf{1.82 \times 10^{-5}}$ | $\mathbf{3.46 \times 10^{-8}}$ | $\mathbf{4.07 \times 10^{-3}}$ | $\mathbf{4.44 \times 10^{-4}}$ | $\mathbf{7.80 \times 10^{-5}}$ | $\mathbf{8.30 \times 10^{-6}}$ | $\mathbf{3.87 \times 10^{-5}}$ | $\mathbf{6.01 \times 10^{-4}}$ |
| 1st Quartile | $\mathbf{3.53 \times 10^{-5}}$ | $\mathbf{4.03 \times 10^{-6}}$ | $\mathbf{5.32 \times 10^{-10}}$ | $\mathbf{8.62 \times 10^{-4}}$ | $\mathbf{5.72 \times 10^{-5}}$ | $\mathbf{3.15 \times 10^{-5}}$ | $\mathbf{6.10 \times 10^{-8}}$ | $\mathbf{3.02 \times 10^{-6}}$ | $\mathbf{1.24 \times 10^{-4}}$ |
| 2nd Quartile | $\mathbf{5.56 \times 10^{-5}}$ | $\mathbf{5.95 \times 10^{-6}}$ | $\mathbf{4.10 \times 10^{-9}}$ | $\mathbf{3.34 \times 10^{-3}}$ | $\mathbf{5.56 \times 10^{-4}}$ | $\mathbf{4.84 \times 10^{-5}}$ | $\mathbf{2.80 \times 10^{-6}}$ | $\mathbf{3.69 \times 10^{-5}}$ | $\mathbf{5.06 \times 10^{-4}}$ |
| 3rd Quartile | $\mathbf{2.82 \times 10^{-4}}$ | $\mathbf{2.99 \times 10^{-5}}$ | $\mathbf{1.06 \times 10^{-8}}$ | $\mathbf{4.23 \times 10^{-3}}$ | $\mathbf{7.20 \times 10^{-4}}$ | $\mathbf{9.22 \times 10^{-5}}$ | $\mathbf{1.14 \times 10^{-5}}$ | $\mathbf{5.81 \times 10^{-5}}$ | $\mathbf{6.78 \times 10^{-4}}$ |
| **MC** | | | | | | | | | |
| Mean | $2.49 \times 10^{-2}$ | $1.83 \times 10^{-3}$ | $1.52 \times 10^{-5}$ | $1.83 \times 10^{-2}$ | $6.71 \times 10^{-3}$ | $2.23 \times 10^{-3}$ | $8.82 \times 10^{-4}$ | $1.76 \times 10^{-3}$ | $7.08 \times 10^{-3}$ |
| 1st Quartile | $3.08 \times 10^{-4}$ | $1.18 \times 10^{-4}$ | $2.33 \times 10^{-6}$ | $4.15 \times 10^{-3}$ | $2.74 \times 10^{-3}$ | $7.89 \times 10^{-4}$ | $1.52 \times 10^{-4}$ | $2.41 \times 10^{-4}$ | $1.06 \times 10^{-3}$ |
| 2nd Quartile | $4.44 \times 10^{-4}$ | $7.84 \times 10^{-4}$ | $6.98 \times 10^{-6}$ | $1.20 \times 10^{-2}$ | $4.97 \times 10^{-3}$ | $1.46 \times 10^{-3}$ | $3.49 \times 10^{-4}$ | $1.24 \times 10^{-3}$ | $2.66 \times 10^{-3}$ |
| 3rd Quartile | $3.69 \times 10^{-3}$ | $2.81 \times 10^{-3}$ | $9.57 \times 10^{-6}$ | $2.38 \times 10^{-2}$ | $7.19 \times 10^{-3}$ | $3.67 \times 10^{-3}$ | $7.33 \times 10^{-4}$ | $2.95 \times 10^{-3}$ | $5.61 \times 10^{-3}$ |
| **WSL** | | | | | | | | | |
| Mean | $6.38 \times 10^{-2}$ | $9.27 \times 10^{-2}$ | $8.60 \times 10^{-2}$ | $1.58 \times 10^{-1}$ | $1.21 \times 10^{-1}$ | $2.25 \times 10^{-1}$ | $2.36 \times 10^{-1}$ | $1.97 \times 10^{-1}$ | $1.47 \times 10^{-1}$ |
| 1st Quartile | $2.84 \times 10^{-2}$ | $8.48 \times 10^{-3}$ | $3.78 \times 10^{-3}$ | $6.64 \times 10^{-2}$ | $5.27 \times 10^{-2}$ | $1.61 \times 10^{-1}$ | $1.44 \times 10^{-2}$ | $6.43 \times 10^{-2}$ | $4.99 \times 10^{-2}$ |
| 2nd Quartile | $4.33 \times 10^{-2}$ | $3.10 \times 10^{-2}$ | $1.43 \times 10^{-2}$ | $1.82 \times 10^{-1}$ | $1.02 \times 10^{-1}$ | $1.87 \times 10^{-1}$ | $1.12 \times 10^{-1}$ | $9.14 \times 10^{-2}$ | $9.55 \times 10^{-2}$ |
| 3rd Quartile | $7.02 \times 10^{-2}$ | $1.00 \times 10^{-1}$ | $1.51 \times 10^{-1}$ | $2.38 \times 10^{-1}$ | $1.69 \times 10^{-1}$ | $2.24 \times 10^{-1}$ | $3.37 \times 10^{-1}$ | $2.08 \times 10^{-1}$ | $1.87 \times 10^{-1}$ |
| **OFA_S** | | | | | | | | | |
| Mean | $5.02 \times 10^{-2}$ | $3.46 \times 10^{-2}$ | $4.74 \times 10^{-2}$ | $4.96 \times 10^{-2}$ | $4.98 \times 10^{-2}$ | $5.03 \times 10^{-2}$ | $5.18 \times 10^{-2}$ | $5.05 \times 10^{-2}$ | $4.80 \times 10^{-2}$ |
| 1st Quartile | $2.53 \times 10^{-2}$ | $2.09 \times 10^{-2}$ | $4.64 \times 10^{-2}$ | $3.68 \times 10^{-2}$ | $4.13 \times 10^{-2}$ | $4.65 \times 10^{-2}$ | $4.29 \times 10^{-2}$ | $4.50 \times 10^{-2}$ | $3.81 \times 10^{-2}$ |
| 2nd Quartile | $5.47 \times 10^{-2}$ | $3.41 \times 10^{-2}$ | $4.90 \times 10^{-2}$ | $4.62 \times 10^{-2}$ | $4.83 \times 10^{-2}$ | $5.02 \times 10^{-2}$ | $5.06 \times 10^{-2}$ | $4.81 \times 10^{-2}$ | $4.76 \times 10^{-2}$ |
| 3rd Quartile | $7.02 \times 10^{-2}$ | $3.91 \times 10^{-2}$ | $5.25 \times 10^{-2}$ | $5.39 \times 10^{-2}$ | $5.81 \times 10^{-2}$ | $5.20 \times 10^{-2}$ | $6.22 \times 10^{-2}$ | $5.28 \times 10^{-2}$ | $5.51 \times 10^{-2}$ |
| **WeightedSHAP** | | | | | | | | | |
| Mean | $3.06 \times 10^{-2}$ | $3.62 \times 10^{-2}$ | $3.74 \times 10^{-2}$ | $9.58 \times 10^{-2}$ | $5.18 \times 10^{-2}$ | $7.77 \times 10^{-2}$ | $8.02 \times 10^{-2}$ | $9.01 \times 10^{-2}$ | $6.25 \times 10^{-2}$ |
| 1st Quartile | $4.26 \times 10^{-3}$ | $5.86 \times 10^{-3}$ | $2.11 \times 10^{-2}$ | $6.57 \times 10^{-2}$ | $3.41 \times 10^{-2}$ | $4.39 \times 10^{-2}$ | $1.70 \times 10^{-2}$ | $2.33 \times 10^{-2}$ | $2.69 \times 10^{-2}$ |
| 2nd Quartile | $1.34 \times 10^{-2}$ | $2.15 \times 10^{-2}$ | $3.27 \times 10^{-2}$ | $8.42 \times 10^{-2}$ | $5.15 \times 10^{-2}$ | $7.77 \times 10^{-2}$ | $1.85 \times 10^{-2}$ | $5.78 \times 10^{-2}$ | $4.47 \times 10^{-2}$ |
| 3rd Quartile | $3.05 \times 10^{-2}$ | $5.82 \times 10^{-2}$ | $4.74 \times 10^{-2}$ | $9.76 \times 10^{-2}$ | $6.13 \times 10^{-2}$ | $1.02 \times 10^{-1}$ | $1.56 \times 10^{-1}$ | $1.37 \times 10^{-1}$ | $8.63 \times 10^{-2}$ |
| **MSR** | | | | | | | | | |
| Mean | $3.06 \times 10^{-2}$ | $2.60 \times 10^{-2}$ | $2.38 \times 10^{-2}$ | $3.56 \times 10^{-2}$ | $2.81 \times 10^{-2}$ | $2.47 \times 10^{-2}$ | $2.27 \times 10^{-2}$ | $2.65 \times 10^{-2}$ | $2.73 \times 10^{-2}$ |
| 1st Quartile | $2.51 \times 10^{-2}$ | $2.00 \times 10^{-2}$ | $1.67 \times 10^{-2}$ | $2.11 \times 10^{-2}$ | $2.44 \times 10^{-2}$ | $2.25 \times 10^{-2}$ | $1.89 \times 10^{-2}$ | $2.27 \times 10^{-2}$ | $2.14 \times 10^{-2}$ |
| 2nd Quartile | $2.66 \times 10^{-2}$ | $2.83 \times 10^{-2}$ | $2.49 \times 10^{-2}$ | $3.84 \times 10^{-2}$ | $2.85 \times 10^{-2}$ | $2.49 \times 10^{-2}$ | $2.19 \times 10^{-2}$ | $2.63 \times 10^{-2}$ | $2.75 \times 10^{-2}$ |
| 3rd Quartile | $3.72 \times 10^{-2}$ | $3.20 \times 10^{-2}$ | $2.99 \times 10^{-2}$ | $4.82 \times 10^{-2}$ | $3.30 \times 10^{-2}$ | $2.73 \times 10^{-2}$ | $2.56 \times 10^{-2}$ | $2.97 \times 10^{-2}$ | $3.29 \times 10^{-2}$ |
| **MSR (paired)** | | | | | | | | | |
| Mean | $4.14 \times 10^{-2}$ | $5.62 \times 10^{-2}$ | $5.11 \times 10^{-2}$ | $6.02 \times 10^{-2}$ | $4.43 \times 10^{-2}$ | $4.62 \times 10^{-2}$ | $4.92 \times 10^{-2}$ | $5.08 \times 10^{-2}$ | $4.99 \times 10^{-2}$ |
| 1st Quartile | $3.09 \times 10^{-2}$ | $3.81 \times 10^{-2}$ | $3.57 \times 10^{-2}$ | $4.27 \times 10^{-2}$ | $4.08 \times 10^{-2}$ | $4.06 \times 10^{-2}$ | $4.34 \times 10^{-2}$ | $4.51 \times 10^{-2}$ | $3.97 \times 10^{-2}$ |
| 2nd Quartile | $4.15 \times 10^{-2}$ | $4.90 \times 10^{-2}$ | $4.16 \times 10^{-2}$ | $6.13 \times 10^{-2}$ | $4.64 \times 10^{-2}$ | $4.48 \times 10^{-2}$ | $5.09 \times 10^{-2}$ | $5.14 \times 10^{-2}$ | $4.84 \times 10^{-2}$ |
| 3rd Quartile | $5.01 \times 10^{-2}$ | $6.03 \times 10^{-2}$ | $5.54 \times 10^{-2}$ | $7.67 \times 10^{-2}$ | $4.74 \times 10^{-2}$ | $4.69 \times 10^{-2}$ | $5.14 \times 10^{-2}$ | $5.51 \times 10^{-2}$ | $5.54 \times 10^{-2}$ |
| **ARM** | | | | | | | | | |
| Mean | $4.22 \times 10^{-2}$ | $3.58 \times 10^{-2}$ | $3.95 \times 10^{-2}$ | $5.16 \times 10^{-2}$ | $4.33 \times 10^{-2}$ | $4.32 \times 10^{-2}$ | $4.38 \times 10^{-2}$ | $3.08 \times 10^{-2}$ | $4.13 \times 10^{-2}$ |
| 1st Quartile | $1.71 \times 10^{-2}$ | $2.36 \times 10^{-2}$ | $2.36 \times 10^{-2}$ | $2.96 \times 10^{-2}$ | $3.06 \times 10^{-2}$ | $2.82 \times 10^{-2}$ | $2.65 \times 10^{-2}$ | $2.55 \times 10^{-2}$ | $2.56 \times 10^{-2}$ |
| 2nd Quartile | $2.19 \times 10^{-2}$ | $3.39 \times 10^{-2}$ | $3.13 \times 10^{-2}$ | $4.48 \times 10^{-2}$ | $3.79 \times 10^{-2}$ | $3.74 \times 10^{-2}$ | $3.16 \times 10^{-2}$ | $2.68 \times 10^{-2}$ | $3.32 \times 10^{-2}$ |
| 3rd Quartile | $4.64 \times 10^{-2}$ | $4.54 \times 10^{-2}$ | $4.84 \times 10^{-2}$ | $5.13 \times 10^{-2}$ | $4.39 \times 10^{-2}$ | $4.81 \times 10^{-2}$ | $5.47 \times 10^{-2}$ | $3.18 \times 10^{-2}$ | $4.62 \times 10^{-2}$ |
| **GELS** | | | | | | | | | |
| Mean | $6.51 \times 10^{-2}$ | $6.07 \times 10^{-2}$ | $8.98 \times 10^{-2}$ | $1.47 \times 10^{-1}$ | $8.48 \times 10^{-2}$ | $7.63 \times 10^{-2}$ | $1.12 \times 10^{-1}$ | $1.05 \times 10^{-1}$ | $9.26 \times 10^{-2}$ |
| 1st Quartile | $4.10 \times 10^{-2}$ | $2.71 \times 10^{-2}$ | $4.01 \times 10^{-2}$ | $6.15 \times 10^{-2}$ | $6.60 \times 10^{-2}$ | $6.37 \times 10^{-2}$ | $5.93 \times 10^{-2}$ | $6.60 \times 10^{-2}$ | $5.31 \times 10^{-2}$ |
| 2nd Quartile | $5.53 \times 10^{-2}$ | $4.13 \times 10^{-2}$ | $5.32 \times 10^{-2}$ | $8.31 \times 10^{-2}$ | $7.63 \times 10^{-2}$ | $7.10 \times 10^{-2}$ | $6.80 \times 10^{-2}$ | $1.05 \times 10^{-1}$ | $6.91 \times 10^{-2}$ |
| 3rd Quartile | $7.15 \times 10^{-2}$ | $7.38 \times 10^{-2}$ | $9.53 \times 10^{-2}$ | $1.33 \times 10^{-1}$ | $9.39 \times 10^{-2}$ | $7.71 \times 10^{-2}$ | $9.54 \times 10^{-2}$ | $1.37 \times 10^{-1}$ | $9.72 \times 10^{-2}$ |
| **AME** | | | | | | | | | |
| Mean | $5.40 \times 10^{-2}$ | $6.39 \times 10^{-2}$ | $4.21 \times 10^{-2}$ | $5.32 \times 10^{-2}$ | $5.34 \times 10^{-2}$ | $5.42 \times 10^{-2}$ | $4.82 \times 10^{-2}$ | $4.89 \times 10^{-2}$ | $5.22 \times 10^{-2}$ |
| 1st Quartile | $3.12 \times 10^{-2}$ | $4.92 \times 10^{-2}$ | $3.53 \times 10^{-2}$ | $4.05 \times 10^{-2}$ | $4.24 \times 10^{-2}$ | $4.70 \times 10^{-2}$ | $4.88 \times 10^{-2}$ | $4.76 \times 10^{-2}$ | $4.28 \times 10^{-2}$ |
| 2nd Quartile | $5.69 \times 10^{-2}$ | $5.76 \times 10^{-2}$ | $4.34 \times 10^{-2}$ | $4.77 \times 10^{-2}$ | $5.24 \times 10^{-2}$ | $5.70 \times 10^{-2}$ | $4.97 \times 10^{-2}$ | $4.89 \times 10^{-2}$ | $5.17 \times 10^{-2}$ |
| 3rd Quartile | $7.39 \times 10^{-2}$ | $7.84 \times 10^{-2}$ | $5.07 \times 10^{-2}$ | $5.96 \times 10^{-2}$ | $6.34 \times 10^{-2}$ | $6.38 \times 10^{-2}$ | $5.01 \times 10^{-2}$ | $5.05 \times 10^{-2}$ | $6.13 \times 10^{-2}$ |

accuracy of the ranking using Spearman's correlation coefficient and Cayley distance (see Appendix F for details).

**Baselines.**   We compare Kernel Banzhaf to several baselines.

- *Monte Carlo (MC)* The standard Monte Carlo estimator approximates each Banzhaf value individually. It operates by sampling each term in the summation with a probability proportional to its respective weight.

- *Weighted Sampling Lift (WSL) (Kwon & Zou, 2022a)* This method employs a Monte Carlo approach where subsets are sampled according to Shapley weights. These samples are subsequently reweighted to produce unbiased estimates of the Banzhaf values.

- *Weighted SHAP (Kwon & Zou, 2022b)* This estimator extends permutation sampling to the Banzhaf context. Random permutations are drawn and reweighted by the Banzhaf value weights to ensure the final estimates remain unbiased.

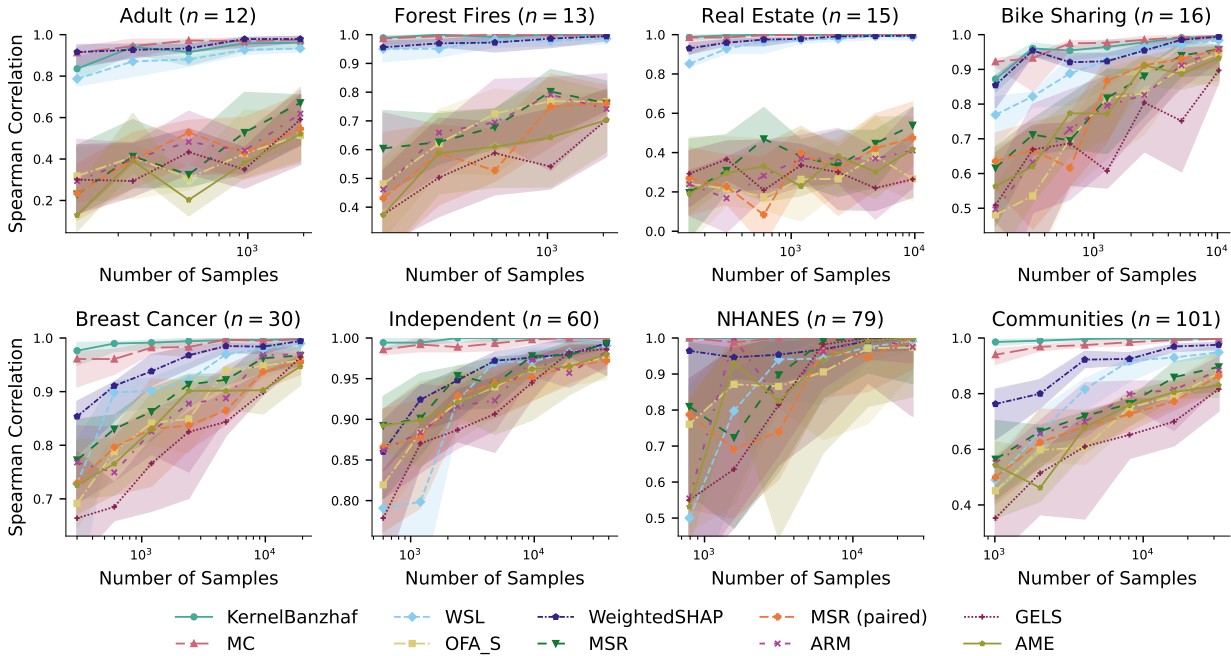

Figure 3: Feature-ranking recovery for non-zero Banzhaf values, measured by Spearman correlation (higher is better). Kernel Banzhaf performs best, particularly on datasets with more features.

- *Maximum Sample Reuse (MSR) Wang & Jia (2023)* MSR is derived from the observation that each Banzhaf value can be decomposed into two distinct summations: one over sets that include player $i$ and one over sets that do not. The estimator computes each summation separately to maximize sample efficiency.

- *MSR (paired)* This variant draws subsets in complement pairs, as in Algorithm 1. It is exactly the direct estimator $\frac{4}{m}\tilde{\mathbf{A}}^\top\tilde{\mathbf{b}}$ that skips the least-squares solve (Appendix H.1).

- *Approximation without Requesting Marginals (ARM) Kolpaczki et al. (2024)* A specialized variant of the MSR estimator, ARM draws half of its samples with probability $p_{|S|-1}$ and the remaining half with probability $p_{|S|}$. To mitigate the numerical instability associated with reweighting, the final estimate selectively includes the first half of samples if $i \in S$ and the second half if $i \notin S$.

- *One-Sample-Fits-All (OFA) Li & Yu (2024b)* Similar to ARM, OFA utilizes the maximum sample reuse principle but samples according to a more complex distribution to optimize estimation.

- *Generic Estimator based on Least Squares (GELS) Li & Yu (2024a)* GELS modifies the estimation process by introducing a dummy variable with a Banzhaf value of 0. Rather than fitting a linear function $f$ iteratively, GELS utilizes the closed-form solution to the regression problem. It applies a sample-reuse Monte Carlo estimator to the underlying matrix-vector multiplication, adjusting the final result by subtracting the value of the dummy variable.

- *Average Marginal Effect (AME) Lin et al. (2022)* AME utilizes an alternative regression formulation. The Banzhaf value is framed as an infinitely tall regression problem, then the estimator samples from this structure to solve an approximate version of the problem.

**Sample Efficiency.** Since evaluating the set function $v$ generally dominates the estimator's time complexity (see confirmation in Appendix D), we investigate estimator error by number of samples in Figure 2. As expected, the error for all estimators decreases as the sample size increases. In terms of performance, Kernel Banzhaf gives the lowest error.

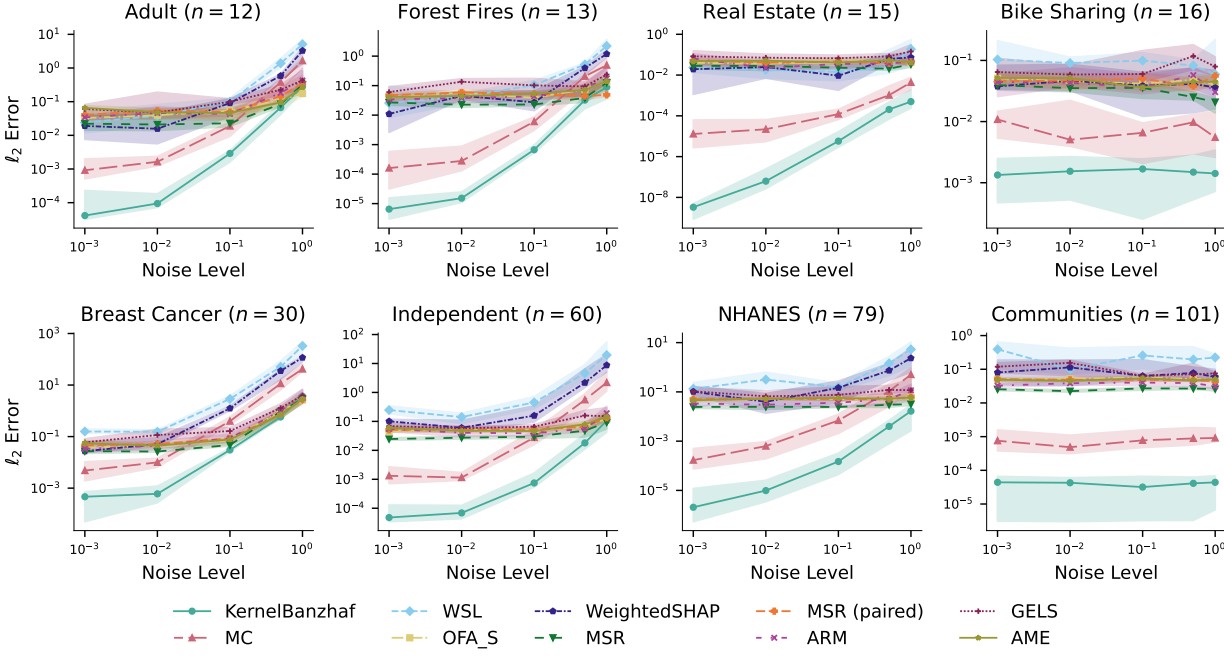

Figure 4: Relative squared $\ell_2$-norm error by noise level across Banzhaf estimators. For each noise level $\sigma$, the estimator observes $v(S) + x$ where $x \sim \mathcal{N}(0, \sigma^2)$. Kernel Banzhaf outperforms the other estimators for lower noise levels or large $n$.

**Robustness to Noise.** The set functions in explainable AI tasks are often estimates of a random process, e.g., the expectation of a model's prediction on a randomly sampled observation. We measure how robust each estimator is to inaccurate set-function values by evaluating the modified function $v(S) + x$, where $x \sim \mathcal{N}(0, \sigma^2)$ and $\sigma$ is the noise level.

As shown in Figure 4, Kernel Banzhaf is more robust to this noise than the other algorithms, especially at low-to-moderate noise levels or when $n$ is large. This suggests that Kernel Banzhaf is particularly well-suited for real-world applications where the set function is approximated. At the highest noise levels, MC and the two MSR variants occasionally match or slightly outperform Kernel Banzhaf. This reversal is consistent with an inversion penalty: conditional on the sampled design $\tilde{\mathbf{A}}$, evaluation noise of variance $\sigma^2$ adds $\sigma^2 \operatorname{tr}[(\tilde{\mathbf{A}}^\top \tilde{\mathbf{A}})^{-1}]$ to Kernel Banzhaf's expected squared error, which is at least $4n\sigma^2/m$ since $\operatorname{tr}(\tilde{\mathbf{A}}^\top \tilde{\mathbf{A}}) = mn/4$ deterministically (AM–HM inequality), with equality only if the sampled Gram matrix is exactly isotropic. MSR, in contrast, does not invert the sampled Gram matrix and incurs added noise error of approximately $4n\sigma^2/m$ with no such penalty. The penalty becomes visible only once noise dominates the total error, and even then the reversal is modest: Kernel Banzhaf's median error remains within a factor of two of both MSR variants, compared with its gains of one to three orders of magnitude at lower noise levels.

**Feature Ranking Recovery.** We also evaluate how accurately each estimator ranks features with non-zero exact Banzhaf values; including zero-valued features would make their relative ranks arbitrary. Figure 3 reports Spearman correlation between the estimated and exact rankings. Kernel Banzhaf gives the most accurate rankings, particularly for larger $n$. Figure 7 in Appendix F corroborates this result using Cayley distance (lower is better).

## 5 Conclusion and Future Work

In this work, we present Kernel Banzhaf, a regression-based approximation algorithm for Banzhaf values. Across eight datasets, Kernel Banzhaf substantially reduces estimation error relative to existing methods. Our theoretical analysis establishes robust approximation guarantees that depend on the size of the Banzhaf values, in contrast to Monte Carlo estimators which depend on the magnitude of the underlying set function.

Appendix B.3 gives an initial evaluation of the extension to probabilistic values. Its accuracy depends on both the weighting and sample budget, motivating a more comprehensive empirical study.

**Subsequent Work.** Since this preprint was originally posted on arXiv, there has been additional work on efficiently approximating Shapley values and Banzhaf values. One successful approach has been Regression MSR (Witter et al., 2025), which builds on regression-based methods, like Kernel Banzhaf, by using them as variance reduction methods for standard Monte Carlo estimators, instead of as direct value estimation algorithms. Indeed, when estimating Banzhaf values, the linear variant of Regression MSR explicitly utilizes Kernel Banzhaf as its surrogate function. Empirically evaluating such recent extensions is outside the scope of this work. Instead, we focus on theoretically and empirically evaluating the base Kernel Banzhaf estimator, which is a necessary prerequisite to understanding its behavior within hybrid frameworks like Regression MSR.

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

# A  Kernel Banzhaf Approximation Guarantees

In this section, we prove theoretical guarantees on the performance of Kernel Banzhaf. We begin with a standard regression sampling result, the proof of which we defer to the end of this section.

**Theorem A.1.** *Consider a subsampling matrix $\mathbf{S} \in \mathbb{R}^{m \times 2^n}$ where each row contains exactly one non-zero entry, selected (possibly in a paired way) with probability proportional to the leverage scores of the corresponding row in $\mathbf{A}$, with the value of the non-zero entry set by the normalization described before Lemma A.3. Define $\tilde{\mathbf{A}} = \mathbf{SA}$ and $\tilde{\mathbf{b}} = \mathbf{Sb}$. Let the solution to the subsampled problem be $\tilde{\boldsymbol{\phi}} = \arg\min_{\mathbf{x}} \|\tilde{\mathbf{A}}\mathbf{x} - \tilde{\mathbf{b}}\|_2^2$. If $m = O(n \log(\frac{n}{\delta}) + \frac{n}{\delta\epsilon})$, then*

$$\|\mathbf{A}\tilde{\boldsymbol{\phi}} - \mathbf{b}\|_2^2 \leq (1 + \epsilon)\|\mathbf{A}\boldsymbol{\phi} - \mathbf{b}\|_2^2 \tag{8}$$

*with probability $1 - \delta$.*

In order to prove Theorem 3.2, we will show that uniform sampling is actually equivalent to leverage score sampling. The leverage scores of a matrix are an important statistical quantity that roughly measures the "uniqueness" of a row (Woodruff et al., 2014; Drineas & Mahoney, 2018).

The leverage scores of $\mathbf{A}$ are given by

$$\ell_S = [\mathbf{A}]_S^\top \left(\mathbf{A}^\top \mathbf{A}\right)^{-1} [\mathbf{A}]_S = \frac{1}{2^{n-2}} \left(\mathbb{1}[S] - \frac{1}{2}\mathbf{1}\right)^\top \left(\mathbb{1}[S] - \frac{1}{2}\mathbf{1}\right) = \frac{n}{2^n} \tag{9}$$

where the second equality follows by Equation 5. Notice the leverage scores are equivalent to uniform sampling, so Theorem A.1 applies directly with uniform sampling in place of leverage score sampling, and Theorem 3.2 immediately follows.

We now use Theorem 3.2 to prove Corollary 3.3: the inequality guaranteed by Theorem 3.2, Equation 6, is exactly the hypothesis the corollary's proof assumes below to derive the $\ell_2$-norm bound on the Banzhaf values themselves.

**Corollary A.2.** *Let $\gamma = \|\mathbf{A}\boldsymbol{\phi} - \mathbf{b}\|_2^2 / \|\mathbf{A}\boldsymbol{\phi}\|_2^2$. Any $\hat{\boldsymbol{\phi}}$ that satisfies Equation 6 also satisfies*

$$\|\hat{\boldsymbol{\phi}} - \boldsymbol{\phi}\|_2^2 \leq \epsilon\gamma\|\boldsymbol{\phi}\|_2^2. \tag{7}$$

*The reverse is also true: Equation 7 implies Equation 6.*

*Proof of Corollary 3.3.* We have

$$\|\mathbf{A}\hat{\boldsymbol{\phi}} - \mathbf{b}\|_2^2 = \|\mathbf{A}\hat{\boldsymbol{\phi}} - \mathbf{A}\boldsymbol{\phi} + \mathbf{A}\boldsymbol{\phi} - \mathbf{b}\|_2^2 = \|\mathbf{A}\hat{\boldsymbol{\phi}} - \mathbf{A}\boldsymbol{\phi}\|_2^2 + \|\mathbf{A}\boldsymbol{\phi} - \mathbf{b}\|_2^2 \tag{10}$$

where the second equality follows because $\mathbf{A}\boldsymbol{\phi} - \mathbf{b}$ is orthogonal to any vector in the span of $\mathbf{A}$. Then, by the assumption that $\|\mathbf{A}\hat{\boldsymbol{\phi}} - \mathbf{b}\|_2^2 \leq (1 + \epsilon)\|\mathbf{A}\boldsymbol{\phi} - \mathbf{b}\|_2^2$, we have $\|\mathbf{A}\hat{\boldsymbol{\phi}} - \mathbf{A}\boldsymbol{\phi}\|_2^2 \leq \epsilon\|\mathbf{A}\boldsymbol{\phi} - \mathbf{b}\|_2^2$. By Equation 5, we have

$$\|\mathbf{A}\boldsymbol{\phi}\|_2^2 = \boldsymbol{\phi}^\top \mathbf{A}^\top \mathbf{A}\boldsymbol{\phi} = 2^{n-2}\|\boldsymbol{\phi}\|_2^2 \tag{11}$$

and, similarly, $\|\mathbf{A}(\hat{\boldsymbol{\phi}} - \boldsymbol{\phi})\|_2^2 = 2^{n-2}\|\hat{\boldsymbol{\phi}} - \boldsymbol{\phi}\|_2^2$. Then, with the definition of $\gamma$,

$$2^{n-2}\|\hat{\boldsymbol{\phi}} - \boldsymbol{\phi}\|_2^2 = \|\mathbf{A}(\hat{\boldsymbol{\phi}} - \boldsymbol{\phi})\|_2^2 \leq \epsilon\|\mathbf{A}\boldsymbol{\phi} - \mathbf{b}\|_2^2 = \epsilon\gamma\|\mathbf{A}\boldsymbol{\phi}\|_2^2 = 2^{n-2}\epsilon\gamma\|\boldsymbol{\phi}\|_2^2. \tag{12}$$

The statement then follows after dividing both sides by $2^{n-2}$. $\qquad\square$

**Comparison to Non-Squared Guarantees.** Corollary 3.3 and the restated Monte Carlo/Maximum Sample Reuse guarantees in Section 3.3 are stated as bounds on the *squared* $\ell_2$-norm error, with sample complexity scaling as $O(1/\epsilon)$. Wang & Jia (2023) instead state their Theorems 4.8 and 4.9 directly on the (non-squared) error, i.e., a bound of the form $\Pr(\|\hat{\phi} - \phi\|_2 \geq \epsilon') \leq \delta$ using $O(n^2/\epsilon'^2 \log(n/\delta))$ samples for MC and $O(n/\epsilon'^2 \log(n/\delta))$ samples for MSR. These are consistent with the guarantees we restate: substituting $\epsilon = \epsilon'^2$ (and, for Kernel Banzhaf, $\epsilon = \epsilon'^2/\gamma$ into Corollary 3.3) converts our squared-error, $O(1/\epsilon)$-sample bounds into non-squared, $O(1/\epsilon'^2)$-sample bounds directly comparable to Wang & Jia (2023)'s Theorems 4.8 and 4.9. Because the same substitution applies uniformly across all three methods, the qualitative comparison between them (a dependence on $\|\phi\|_2^2$ and $\gamma$ for Kernel Banzhaf versus a dependence on $\max_S v(S)^2$ for MC/MSR) holds regardless of which convention is used.

The factors of $\|\phi\|_2^2$ appearing in $\gamma$ and in Corollary 3.3's bound in fact cancel: since $\|\mathbf{A}\phi\|_2^2 = 2^{n-2}\|\phi\|_2^2$ (Equation 5), $\gamma\|\phi\|_2^2 = \|\mathbf{A}\phi - \mathbf{b}\|_2^2/2^{n-2}$, so Corollary 3.3 can equivalently be stated purely in terms of the regression residual: with $m = O(n\log(n/\delta) + n/(\delta\epsilon))$ samples, $\|\hat{\phi} - \phi\|_2^2 \leq \epsilon\|\mathbf{A}\phi - \mathbf{b}\|_2^2/2^{n-2}$ with probability $1 - \delta$. Converting to the non-squared convention as above gives $m = O\left(n\log(n/\delta) + \frac{n}{\delta\epsilon'^2} \cdot \frac{\|\mathbf{A}\phi - \mathbf{b}\|_2^2}{2^{n-2}}\right)$. If in addition $|v(S)| \leq 1$ for every $S$ (the setting of Wang & Jia (2023)), then, since $\phi$ minimizes $\|\mathbf{A}\mathbf{x} - \mathbf{b}\|_2^2$ over $\mathbf{x}$, comparing to $\mathbf{x} = \mathbf{0}$ gives $\|\mathbf{A}\phi - \mathbf{b}\|_2^2 \leq \|\mathbf{b}\|_2^2 \leq 2^n$, and the complexity simplifies to $O(n\log(n/\delta) + n/(\delta\epsilon'^2))$, directly comparable to Wang & Jia (2023)'s form rather than only up to a $\gamma$ prefactor. The boundedness assumption plays the same normalizing role here as in the Monte Carlo and Maximum Sample Reuse bounds, whose $\max_S v(S)^2$ factor it likewise reduces to a constant. Equivalently, $\gamma\|\phi\|_2^2 = 4\,\mathbb{E}_S\left[(v(S) - [\mathbf{A}\phi]_S)^2\right]$ for $S$ drawn uniformly from all subsets: four times the mean squared residual of the best linear approximation of $v$, an expectation over subsets. Without any boundedness assumption, this quantity is at most $4\max_S v(S)^2$, so Corollary 3.3's data-dependent factor never exceeds a constant multiple of the one in the Monte Carlo and Maximum Sample Reuse guarantees, and it is far smaller when $v$ is close to additive.

Next, we prove Theorem A.1, which is a standard guarantee for leverage score sampling. However, because rows are sampled in *pairs*, we need to modify the analysis. In particular, both the spectral guarantee that the sampling matrix preserves eigenvalues and the Frobenius guarantee that the sampling matrix preserves Frobenius norm need to be reproved. Our analysis adapts the leverage score framework used for regression-based Shapley value estimation by Musco & Witter (2025); the new ingredient is the analysis of i.i.d. block sampling, following Wu (2018), which handles rows drawn in identical pairs.

We will adopt the notation from Wu (2018). Let's consider a leverage score sampling method where rows are selected in blocks. Define $\Theta$ as a partition of blocks, each containing 2 elements with identical leverage scores in our setting. Define $p_k := \ell_k/\sum_j \ell_j$ as the (unblocked) normalized leverage score of an individual row $k$. We assign a sampling probability $p_i^+$ to each block $\Theta_i$, calculated as the sum of leverage scores in that block divided by the total sum of all leverage scores: $p_i^+ := \sum_{k \in \Theta_i} p_k = \frac{\sum_{k \in \Theta_i} \ell_k}{\sum_j \ell_j}$. For simplicity of notation, suppose $m$ is even. Let $\mathbf{S} \in \mathbb{R}^{m \times \rho}$ be a sampling matrix, initially set to $\mathbf{0}_{m \times \rho}$. The sampling process repeats $m/2$ times: Sample a block $\Theta_i$ with probability $p_i^+$. For each $k \in \Theta_i$, set the $k$th entry in an empty row to $\frac{1}{\sqrt{(m/2)\,p_i^+}}$, matching the normalization in Proposition 2.2 of Wu (2018) for $m/2$ i.i.d. block draws. We index the resulting $m$ sampled rows directly by $j \in [m]$ below, one per block-draw repeated twice.

To analyze the solution obtained from this block-wise leverage score sampling, we will demonstrate that the sampling matrix $\mathbf{S}$ preserves both the spectral norm and the Frobenius norm.

**Lemma A.3** (Spectral Approximation). *Let $\mathbf{U} \in \mathbb{R}^{\rho \times n}$ be a matrix with orthonormal columns. Consider the block random sampling matrix $\mathbf{S}$ described above with rows sampled according to the leverage scores of $\mathbf{U}$. Let $\epsilon \in (0, 1]$. When $m = \Omega(n\log(n/\delta)/\epsilon^2)$,*

$$\|\mathbf{I} - \mathbf{U}^\top \mathbf{S}^\top \mathbf{S} \mathbf{U}\|_2 \leq \epsilon \tag{13}$$

*with probability $1 - \delta$.*

*Proof of Lemma A.3.* We will use the following matrix Chernoff bound (see e.g., Fact 1 in Woodruff et al. (2014)).

**Fact A.4** (Matrix Chernoff). *Let $\mathbf{X}_1, \ldots, \mathbf{X}_m \in \mathbb{R}^{n \times n}$ be independent samples of symmetric random matrices with $\mathbb{E}[\mathbf{X}_j] = \mathbf{0}$, $\|\mathbf{X}_j\|_2 \leq \gamma$ for all $j$, and $\|\mathbb{E}_j[\mathbf{X}_j^2]\|_2 \leq \sigma^2$. Then for any $\epsilon > 0$,*

$$\Pr\left(\left\|\frac{1}{m}\sum_{j=1}^m \mathbf{X}_j\right\|_2 \geq \epsilon\right) \leq 2n\exp\left(\frac{-m\epsilon^2}{2\sigma^2 + 2\gamma\epsilon/3}\right). \tag{14}$$

For sample $j \in [m]$, let $i(j)$ be the index of the block selected. Define

$$\mathbf{X}_j = \mathbf{I} - \frac{1}{p_{i(j)}^+}\sum_{k \in \Theta_{i(j)}} \mathbf{U}_k^\top \mathbf{U}_k \tag{15}$$

We will compute $\mathbf{E}[X_j]$, $\|\mathbf{X}_j\|_2$, and $\|\mathbb{E}[\mathbf{X}_j^2]\|_2$. First,

$$\mathbb{E}[\mathbf{X}_j] = \mathbf{I} - \sum_{i=1}^{|\Theta|} p_i^+ \frac{1}{p_i^+}\sum_{k \in \Theta_i} \mathbf{U}_k^\top \mathbf{U}_k = \mathbf{0} \tag{16}$$

where the last equality follows because $\Theta$ is a partition and $\mathbf{U}^\top \mathbf{U} = \mathbf{I}$. Next, note that

$$\|\mathbf{X}_j\|_2 \leq \|\mathbf{I}\|_2 + \frac{\sum_{k \in \Theta_{i(j)}} \|\mathbf{U}_k^\top \mathbf{U}_k\|_2}{\sum_{k \in \Theta_{i(j)}} p_k} = 1 + n\frac{\sum_{k \in \Theta_{i(j)}} \|\mathbf{U}_k\|_2^2}{\sum_{k \in \Theta_{i(j)}} \ell_k} = 1 + n \tag{17}$$

where the last equality follows because $\|\mathbf{U}_k\|_2^2 = \ell_k$ since $\mathbf{U}^\top \mathbf{U} = \mathbf{I}$. Define $\mathbf{U}_{\Theta_i} \in \mathbb{R}^{|\Theta_i| \times n}$ as the matrix with rows $\mathbf{U}_k$ for $k \in \Theta_i$. Observe that $\sum_{k \in \Theta_i} \mathbf{U}_k^\top \mathbf{U}_k = \mathbf{U}_{\Theta_i}^\top \mathbf{U}_{\Theta_i}$. Finally, note that

$$\mathbb{E}[\mathbf{X}_j^2] = \mathbf{I} - 2\sum_{i=1}^{|\Theta|} p_i^+ \frac{\sum_{k \in \Theta_i} \mathbf{U}_k^\top \mathbf{U}_k}{p_i^+} + \sum_{i=1}^{|\Theta|} p_i^+ \frac{\left(\mathbf{U}_{\Theta_i}^\top \mathbf{U}_{\Theta_i}\right)^2}{p_i^{+2}} \tag{18}$$

$$= -\mathbf{I} + \sum_{i=1}^{|\Theta|} \frac{1}{p_i^+}\mathbf{U}_{\Theta_i}^\top \mathbf{U}_{\Theta_i}\mathbf{U}_{\Theta_i}^\top \mathbf{U}_{\Theta_i}. \tag{19}$$

Observe that entry $(k, k')$ of $\mathbf{U}_{\Theta_i}\mathbf{U}_{\Theta_i}^\top \in \mathbb{R}^{|\Theta_i| \times |\Theta_i|}$ is $\mathbf{U}_k\mathbf{U}_{k'}^\top$. So the absolute value of each entry is $|\mathbf{U}_k\mathbf{U}_{k'}^\top| \leq \|\mathbf{U}_k\|_2\|\mathbf{U}_{k'}\|_2 = \ell_k^{1/2}\ell_{k'}^{1/2}$ by Cauchy-Schwarz. Define $\ell_i^{\max} = \max_{k \in \Theta_i} \ell_k$ and $\ell_i^{\min} = \min_{k \in \Theta_i} \ell_k$. By the Gershgorin circle theorem, $\mathbf{U}_{\Theta_i}\mathbf{U}_{\Theta_i}^\top \preceq \ell_i^{\max}|\Theta_i|\mathbf{I}$, since each row of $\mathbf{U}_{\Theta_i}\mathbf{U}_{\Theta_i}^\top$ has diagonal entry at most $\ell_i^{\max}$ and the sum of the (at most $|\Theta_i| - 1$) off-diagonal entries in that row is also at most $(|\Theta_i| - 1)\ell_i^{\max}$. Equivalently, $\mathbf{A} \preceq \mathbf{B}$ means that $\mathbf{x}^\top \mathbf{A}\mathbf{x} \leq \mathbf{x}^\top \mathbf{B}\mathbf{x}$ for all $\mathbf{x}$. Consider an arbitrary $\mathbf{z}$. We have $\mathbf{z}^\top \mathbf{C}^\top \mathbf{A}\mathbf{C}\mathbf{z} \leq \mathbf{z}^\top \mathbf{C}^\top \mathbf{B}\mathbf{C}\mathbf{z}$ since $\mathbf{C}\mathbf{z}$ is some $\mathbf{x}$. It follows that $\mathbf{U}_{\Theta_i}^\top \mathbf{U}_{\Theta_i}\mathbf{U}_{\Theta_i}^\top \mathbf{U}_{\Theta_i} \preceq \ell_i^{\max}|\Theta_i|\mathbf{U}_{\Theta_i}^\top \mathbf{U}_{\Theta_i}$. Then, since $\sum_{k \in \Theta_i} \ell_k \geq |\Theta_i|\ell_i^{\min}$, the factor of $|\Theta_i|$ in the numerator below cancels against this lower bound on the denominator, and

$$(19) \preceq n\sum_{i=1}^{|\Theta|} \frac{\ell_i^{\max}|\Theta_i|\mathbf{U}_{\Theta_i}^\top \mathbf{U}_{\Theta_i}}{\sum_{k \in \Theta_i} \ell_k} \preceq n\max_i \frac{\ell_i^{\max}}{\ell_i^{\min}}\mathbf{I}. \tag{20}$$

Since the leverage scores in a block are all equal, $\|\mathbb{E}[\mathbf{X}_j^2]\|_2 \leq n$.

The matrices $\mathbf{X}_1, \ldots, \mathbf{X}_m$ are not themselves independent: the two rows produced by each block draw carry the same block label, so $\mathbf{X}_{2t-1} = \mathbf{X}_{2t}$ for every draw $t \in [m/2]$. However, the $m/2$ per-draw matrices are independent, and, because each is repeated exactly twice, $\frac{1}{m}\sum_{j=1}^m \mathbf{X}_j = \frac{1}{m/2}\sum_{t=1}^{m/2} \mathbf{X}_{2t}$. Applying Fact A.4 to the $m/2$ independent per-draw matrices, with $m = \Omega(n\log(n/\delta)/\epsilon^2)$, yields

$$\Pr\left(\left\|\frac{1}{m}\sum_{j=1}^m \left(\mathbf{I} - \frac{1}{p_{i(j)}^+}\mathbf{U}_{\Theta_{i(j)}}^\top \mathbf{U}_{\Theta_{i(j)}}\right)\right\|_2 \geq \epsilon\right) \leq \delta. \tag{21}$$

(Relative to applying the fact directly to $m$ independent samples, the exponent carries $m/2$ in place of $m$; this factor of 2 is absorbed into the constant of the $\Omega(\cdot)$ requirement on $m$.) The lemma statement follows.

$\square$

We will also show that the sampling matrix preserves the Frobenius norm.

**Lemma A.5** (Frobenius Approximation). *Let* $\mathbf{U} \in \mathbb{R}^{\rho \times n}$ *be a matrix with orthonormal columns. Consider the block random sampling matrix* $\mathbf{S}$ *described above with rows sampled according to the leverage scores of* $\mathbf{U}$. *Let* $\mathbf{V} \in \mathbb{R}^{\rho \times n'}$. *As long as* $m \geq \frac{2}{\delta \epsilon^2}$, *then*

$$\left\| \mathbf{U}^\top \mathbf{S}^\top \mathbf{S} \mathbf{V} - \mathbf{U}^\top \mathbf{V} \right\|_F \leq \epsilon \|\mathbf{U}\|_F \|\mathbf{V}\|_F \tag{22}$$

*with probability* $1 - \delta$.

*Proof of Lemma A.5.* By Proposition 2.2 in Wu (2018), applied with $c = m/2$ i.i.d. block draws, we have that

$$\mathbb{E}[\|\mathbf{U}^\top \mathbf{S}^\top \mathbf{S} \mathbf{V} - \mathbf{U}^\top \mathbf{V}\|_F^2] + \frac{2}{m} \|\mathbf{U}^\top \mathbf{V}\|_F^2 = \frac{2}{m} \sum_{i=1}^{|\Theta|} \frac{1}{p_i^+} \left\| \mathbf{U}_{\Theta_i}^\top \mathbf{V}_{\Theta_i} \right\|_F^2, \tag{23}$$

where, as in the proof of Lemma A.3, $\mathbf{U}_{\Theta_i} \in \mathbb{R}^{|\Theta_i| \times n}$ is the matrix with rows $\mathbf{U}_k$ for $k \in \Theta_i$, and $\mathbf{V}_{\Theta_i} \in \mathbb{R}^{|\Theta_i| \times n'}$ is defined analogously. Dropping the non-negative term on the left and applying submultiplicativity, $\|\mathbf{U}_{\Theta_i}^\top \mathbf{V}_{\Theta_i}\|_F \leq \|\mathbf{U}_{\Theta_i}\|_F \|\mathbf{V}_{\Theta_i}\|_F$, gives

$$\mathbb{E}[\|\mathbf{U}^\top \mathbf{S}^\top \mathbf{S} \mathbf{V} - \mathbf{U}^\top \mathbf{V}\|_F^2] \leq \frac{2}{m} \sum_{i=1}^{|\Theta|} \frac{1}{p_i^+} \|\mathbf{U}_{\Theta_i}\|_F^2 \|\mathbf{V}_{\Theta_i}\|_F^2. \tag{24}$$

Because $\mathbf{U}$ has orthonormal columns, $\ell_k = \|\mathbf{U}_k\|_2^2$, so $p_k = \|\mathbf{U}_k\|_2^2 / \|\mathbf{U}\|_F^2$ by the definition of leverage scores; we have

$$\mathbb{E}[\|\mathbf{U}^\top \mathbf{S}^\top \mathbf{S} \mathbf{V} - \mathbf{U}^\top \mathbf{V}\|_F^2] \leq \frac{2}{m} \sum_{i=1}^{|\Theta|} \|\mathbf{U}\|_F^2 \frac{\sum_{k \in \Theta_i} \|\mathbf{U}_k\|_2^2}{\sum_{k \in \Theta_i} \|\mathbf{U}_k\|_2^2} \|\mathbf{V}_{\Theta_i}\|_F^2 = \frac{2}{m} \|\mathbf{U}\|_F^2 \|\mathbf{V}\|_F^2. \tag{25}$$

By Markov's inequality,

$$\Pr\left( \left\| \mathbf{U}^\top \mathbf{S}^\top \mathbf{S} \mathbf{V} - \mathbf{U}^\top \mathbf{V} \right\|_F > \epsilon \|\mathbf{U}\|_F \|\mathbf{V}\|_F \right) \leq \frac{\mathbb{E}\left[ \left\| \mathbf{U}^\top \mathbf{S}^\top \mathbf{S} \mathbf{V} - \mathbf{U}^\top \mathbf{V} \right\|_F^2 \right]}{\epsilon^2 \|\mathbf{U}\|_F^2 \|\mathbf{V}\|_F^2} \leq \frac{2}{m\epsilon^2} \leq \delta \tag{26}$$

as long as $m \geq \frac{2}{\delta \epsilon^2}$. $\qquad\square$

With Lemmas A.3 and A.5 already proved for the special paired leverage score sampling, the following analysis is standard. We include the following proof for completeness.

*Proof of Theorem A.1.* Observe that

$$\|\mathbf{A}\tilde{\phi} - \mathbf{b}\|_2^2 = \|\mathbf{A}\tilde{\phi} - \mathbf{A}\phi + \mathbf{A}\phi - \mathbf{b}\|_2^2 = \|\mathbf{A}\tilde{\phi} - \mathbf{A}\phi\|_2^2 + \|\mathbf{A}\phi - \mathbf{b}\|_2^2 \tag{27}$$

where the second equality follows because $\mathbf{A}\phi - \mathbf{b}$ is orthogonal to any vector in the span of $\mathbf{A}$. So to prove the theorem, it suffices to show that

$$\|\mathbf{A}\tilde{\phi} - \mathbf{A}\phi\|_2^2 \leq \epsilon \|\mathbf{A}\phi - \mathbf{b}\|_2^2. \tag{28}$$

Let $\mathbf{U} \in \mathbb{R}^{\rho \times n}$ be an orthonormal matrix that spans the columns of $\mathbf{A}$. There is some $\mathbf{y}$ such that $\mathbf{U}\mathbf{y} = \mathbf{A}\phi$ and some $\tilde{\mathbf{y}}$ such that $\mathbf{U}\tilde{\mathbf{y}} = \mathbf{A}\tilde{\phi}$. Observe that $\|\mathbf{A}\tilde{\phi} - \mathbf{A}\phi\|_2 = \|\mathbf{U}\tilde{\mathbf{y}} - \mathbf{U}\mathbf{y}\|_2 = \|\tilde{\mathbf{y}} - \mathbf{y}\|_2$ where the last equality follows because $\mathbf{U}^\top \mathbf{U} = \mathbf{I}$.

By the reverse triangle inequality and the submultiplicavity of the spectral norm, we have

$$\|\tilde{\mathbf{y}} - \mathbf{y}\|_2 \leq \|\mathbf{U}^\top \mathbf{S}^\top \mathbf{S} \mathbf{U}(\tilde{\mathbf{y}} - \mathbf{y})\|_2 + \|\mathbf{U}^\top \mathbf{S}^\top \mathbf{S} \mathbf{U}(\tilde{\mathbf{y}} - \mathbf{y}) - (\tilde{\mathbf{y}} - \mathbf{y})\|_2 \tag{29}$$

$$\leq \|\mathbf{U}^\top \mathbf{S}^\top \mathbf{S} \mathbf{U}(\tilde{\mathbf{y}} - \mathbf{y})\|_2 + \|\mathbf{U}^\top \mathbf{S}^\top \mathbf{S} \mathbf{U} - \mathbf{I}\|_2 \|\tilde{\mathbf{y}} - \mathbf{y}\|_2. \tag{30}$$

Because $\mathbf{U}$ has the same leverage scores as $\mathbf{A}$, we can apply Lemma A.3: With $m = O(n \log \frac{n}{\delta})$, we have $\|\mathbf{U}^\top \mathbf{S}^\top \mathbf{S} \mathbf{U} - \mathbf{I}\|_2 \leq \frac{1}{2}$ with probability $1 - \delta/2$. So, with probability $1 - \delta/2$,

$$\|\tilde{\mathbf{y}} - \mathbf{y}\|_2 \leq 2\|\mathbf{U}^\top \mathbf{S}^\top \mathbf{S} \mathbf{U}(\tilde{\mathbf{y}} - \mathbf{y})\|_2. \tag{31}$$

Then

$$\|\mathbf{U}^\top \mathbf{S}^\top \mathbf{S} \mathbf{U}(\tilde{\mathbf{y}} - \mathbf{y})\|_2 = \left\|\mathbf{U}^\top \mathbf{S}^\top \left(\mathbf{S}\mathbf{U}\tilde{\mathbf{y}} - \mathbf{S}\mathbf{b} + \mathbf{S}\mathbf{b} - \mathbf{S}\mathbf{U}\mathbf{y}\right)\right\|_2 \tag{32}$$

$$= \left\|\mathbf{U}^\top \mathbf{S}^\top \mathbf{S}\left(\mathbf{U}\mathbf{y} - \mathbf{b}\right)\right\|_2 \tag{33}$$

where the second equality follows because $\mathbf{S}\mathbf{U}\tilde{\mathbf{y}} - \mathbf{S}\mathbf{b}$ is orthogonal to any vector in the span of $\mathbf{S}\mathbf{U}$. By similar reasoning, notice that $\mathbf{U}^\top(\mathbf{U}\mathbf{y} - \mathbf{b}) = \mathbf{0}$. Then, as long as $m = O(\frac{n}{\delta\epsilon})$, we have

$$\left\|\mathbf{U}^\top \mathbf{S}^\top \mathbf{S}\left(\mathbf{U}\mathbf{y} - \mathbf{b}\right)\right\|_2 \leq \frac{\sqrt{\epsilon}}{2\sqrt{n}}\|\mathbf{U}\|_F\|\mathbf{U}\mathbf{y} - \mathbf{b}\|_2 \tag{34}$$

with probability $1 - \delta/2$ by Lemma A.5. Since $\mathbf{U}$ has orthonormal columns, $\|\mathbf{U}\|_F^2 \leq n$. Then, combining inequalities yields

$$\|\mathbf{A}\tilde{\boldsymbol{\phi}} - \mathbf{A}\boldsymbol{\phi}\|_2^2 = \|\tilde{\mathbf{y}} - \mathbf{y}\|_2^2 \leq 4\|\mathbf{U}^\top \mathbf{S}^\top \mathbf{S} \mathbf{U}(\tilde{\mathbf{y}} - \mathbf{y})\|_2^2 \leq \epsilon\|\mathbf{U}\mathbf{y} - \mathbf{b}\|_2^2 = \epsilon\|\mathbf{A}\boldsymbol{\phi} - \mathbf{b}\|_2^2 \tag{35}$$

with probability $1 - \delta$, where the first inequality squares both sides of Equation 31, and the second squares both sides of the bound obtained by combining Equation 34 with $\|\mathbf{U}\|_F^2 \leq n$. $\qquad\square$

## B  Extension to Probabilistic Values

Banzhaf and Shapley values belong to the broader class of *probabilistic values*, which also arise in machine-learning applications (Kwon & Zou, 2022a; Li & Yu, 2024b;c). We extend the regression formulation from Section 3.1 to this class and report an exploratory empirical evaluation in Appendix B.3.

In this section, we first establish how the optimal solution connects to the probabilistic values. Next, we prove guarantees on the approximate solution selected via leverage score sampling.

For a weight vector $\mathbf{p} \in [0,1]^n$ such that $\sum_{\ell=0}^{n-1} \binom{n-1}{\ell} p_\ell = 1$, the *probabilistic value* is given by

$$\phi_i^{\mathrm{prob}} = \sum_{S \subseteq [n] \setminus \{i\}} p_{|S|} [v(S \cup \{i\}) - v(S)].$$

Define the quantities $a_n$ and $b_n$ as follows:

$$a_n := 2 \sum_{\ell=0}^{n-1} \binom{n-1}{\ell} p_\ell^2 - \sum_{\ell=1}^{n-1} \binom{n-2}{\ell-1} (p_\ell - p_{\ell-1})^2 \tag{36}$$

$$b_n := \frac{1}{a_n} \sum_{\ell=1}^{n-1} \binom{n-2}{\ell-1} (p_\ell - p_{\ell-1})^2. \tag{37}$$

Note that both $a_n$ and $b_n$ depend only on $n$ and $\mathbf{p}$, and can be computed in $O(n)$ time. In the Banzhaf setting when $p_\ell = p_{\ell-1}$ for all $\ell$, observe that $b_n = 0$.

In the remainder of this section, let $\mathbf{A} \in \mathbb{R}^{2^n \times n}$ such that

$$[\mathbf{A}]_{S,i} = \begin{cases} \frac{p_{|S|-1}}{a_n} & \text{if } i \in S \\ -\frac{p_{|S|}}{a_n} & \text{if } i \notin S \end{cases} \tag{38}$$

where $i \in [n]$ and we use $S \subseteq [n]$ as an index for ease of notation. Let $\mathbf{b} \in \mathbb{R}^{2^n}$ be the vector representation of the set function $v$ i.e., $b_S = v(S)$.

### B.1  Equivalence to Probabilistic Values

Like for Banzhaf values, let $\mathbf{b}_S^{\mathrm{prob}} = v(S)$, using $\mathbf{A}^{\mathrm{prob}}$ to denote $\mathbf{A}$ above when we wish to emphasize the dependence on $\mathbf{p}$.

**Lemma B.1** (Extended Regression Equivalence). *Let*

$$\mathbf{x}^* = \arg\min_{\mathbf{x} \in \mathbb{R}^n} \|\mathbf{A}^{\mathrm{prob}} \mathbf{x} - \mathbf{b}^{\mathrm{prob}}\|_2^2.$$

*Then* $\boldsymbol{\phi}^{\mathrm{prob}} = (\mathbf{I} + b_n \mathbf{1}\mathbf{1}^\top) \mathbf{x}^*$.

*Proof of Lemma B.1.* Our proof strategy is to explicitly compute the optimal solution

$$\mathbf{x}^* = (\mathbf{A}^\top \mathbf{A})^{-1} \mathbf{A}^\top \mathbf{b}. \tag{39}$$

We begin with $\mathbf{A}^\top \mathbf{A}$. In the computation, we will repeatedly use the symmetry of $\mathbf{A}^\top \mathbf{A}$; namely, all the diagonal entries are the same and all the non-diagonal entries are the same. The diagonal entries for $i \in [n]$ are given by

$$a_n^2 [\mathbf{A}^\top \mathbf{A}]_{i,i} = \sum_{S \subseteq [n]: i \in S} p_{|S|-1}^2 + \sum_{S \subseteq [n]: i \notin S} p_{|S|}^2 = \sum_{\ell=1}^{n} \binom{n-1}{\ell-1} p_{\ell-1}^2 + \sum_{\ell=0}^{n-1} \binom{n-1}{\ell} p_\ell^2 \tag{40}$$

$$= 2 \sum_{\ell=0}^{n-1} \binom{n-1}{\ell} p_\ell^2 \tag{41}$$

where the last equality follows by a change of variable. Similarly, the off-diagonal entries for $i \neq j$ are given by

$$a_n^2[\mathbf{A}^\top\mathbf{A}]_{i,j} = \sum_{S\subseteq[n]:i,j\in S} p_{|S|-1}^2 - 2\sum_{S\subseteq[n]:i\in S,j\notin S} p_{|S|-1}p_{|S|} + \sum_{S\subseteq[n]:i,j\notin S} p_{|S|}^2 \tag{42}$$

$$= \sum_{\ell=2}^{n}\binom{n-2}{\ell-2}p_{\ell-1}^2 - 2\sum_{\ell=1}^{n-1}\binom{n-2}{\ell-1}p_{\ell-1}p_\ell + \sum_{\ell=0}^{n-2}\binom{n-2}{\ell}p_\ell^2 \tag{43}$$

$$= \sum_{\ell=1}^{n-1}\binom{n-2}{\ell-1}(p_\ell - p_{\ell-1})^2 = a_n b_n. \tag{44}$$

By the definition of $a_n$, we have

$$a_n = 2\sum_{\ell=0}^{n-1}\binom{n-1}{\ell}p_\ell^2 - \sum_{\ell=1}^{n-1}\binom{n-2}{\ell-1}(p_\ell - p_{\ell-1})^2 = a_n^2[\mathbf{A}^\top\mathbf{A}]_{i,i} - a_n^2[\mathbf{A}^\top\mathbf{A}]_{i,j}. \tag{45}$$

It follows that $[\mathbf{A}^\top\mathbf{A}]_{i,i} - [\mathbf{A}^\top\mathbf{A}]_{i,j} = \frac{1}{a_n}$. Then

$$\mathbf{A}^\top\mathbf{A} = ([\mathbf{A}^\top\mathbf{A}]_{i,i} - [\mathbf{A}^\top\mathbf{A}]_{i,j})\mathbf{I} + [\mathbf{A}^\top\mathbf{A}]_{i,j}\mathbf{1}\mathbf{1}^\top = \frac{1}{a_n}(\mathbf{I} + b_n\mathbf{1}\mathbf{1}^\top). \tag{46}$$

By the Sherman-Morrison formula, $(\mathbf{I} + b_n\mathbf{1}\mathbf{1}^\top)^{-1} = \mathbf{I} - \frac{b_n}{1+nb_n}\mathbf{1}\mathbf{1}^\top$. Then

$$(\mathbf{A}^\top\mathbf{A})^{-1} = a_n\left(\mathbf{I} - \frac{b_n}{1+nb_n}\mathbf{1}\mathbf{1}^\top\right). \tag{47}$$

Next, we compute $\mathbf{A}^\top\mathbf{v}$. The $i$th entry of this $n\times 1$ vector is given by

$$a_n[\mathbf{A}^\top\mathbf{v}]_i = \sum_{S\subseteq[n]:i\in S} p_{|S|-1}v(S) - \sum_{S\subseteq[n]:i\notin S} p_{|S|}v(S) = \sum_{S\subseteq[n]\backslash\{i\}} p_{|S|}(v(S\cup\{i\}) - v(S)) = \phi_i. \tag{48}$$

Therefore $\mathbf{A}^\top\mathbf{v} = \frac{1}{a_n}\boldsymbol{\phi}$. Finally,

$$\mathbf{x}^* = (\mathbf{A}^\top\mathbf{A})^{-1}\mathbf{A}^\top\mathbf{b} = a_n\left(\mathbf{I} - \frac{b_n}{1+nb_n}\mathbf{1}\mathbf{1}^\top\right)\frac{1}{a_n}\boldsymbol{\phi}$$

and, taking the inverse,

$$\boldsymbol{\phi} = \left(\mathbf{I} - \frac{b_n}{1+nb_n}\mathbf{1}\mathbf{1}^\top\right)^{-1}\mathbf{x}^* = (\mathbf{I} + b_n\mathbf{1}\mathbf{1}^\top)\mathbf{x}^*. \tag{49}$$

$\square$

## B.2 Approximation Guarantees

Like in the Banzhaf setting, we can sample the full regression problem to produce a subsampled design matrix $\tilde{\mathbf{A}}^{\mathrm{prob}} \in \mathbb{R}^{m\times n}$ and target vector $\tilde{\mathbf{b}}^{\mathrm{prob}} \in \mathbb{R}^m$. Let $\tilde{\boldsymbol{\phi}}^{\mathrm{prob}} = \arg\min_\mathbf{x}\|\tilde{\mathbf{A}}^{\mathrm{prob}}\mathbf{x} - \tilde{\mathbf{b}}^{\mathrm{prob}}\|_2^2$ be the solution to this subsampled problem. The following theorem gives guarantees on the accuracy of this solution.

**Theorem B.2** (Probabilistic Value Approximation). *Let $\gamma = \|\mathbf{A}^{\mathrm{prob}}\mathbf{x}^* - \mathbf{b}^{\mathrm{prob}}\|_2^2/\|\mathbf{A}^{\mathrm{prob}}\mathbf{x}^*\|_2^2$. There is a sampling method that uses $m = O(n\log\frac{n}{\delta} + \frac{n}{\delta\epsilon})$ samples to compute an estimate, $\tilde{\boldsymbol{\phi}}^{\mathrm{prob}}$, such that, with probability $1 - \delta$,*

$$\|\boldsymbol{\phi}^{\mathrm{prob}} - \tilde{\boldsymbol{\phi}}^{\mathrm{prob}}\|_2^2 \leq \epsilon\gamma(1+nb_n)^3\|\boldsymbol{\phi}^{\mathrm{prob}}\|_2^2.$$

Theorem B.2 recovers Corollary 3.3 for Banzhaf values because $b_n = 0$. Appendix B.3 evaluates nine weightings, including Beta Shapley (Kwon & Zou, 2022a). At $m = 40n$, the regression estimator is most accurate for the near-uniform and two most asymmetric weightings, whereas direct estimators are stronger for the intermediate weightings. The full budget sweep shows that the asymmetric cases favor the regression estimator only from moderate budgets onward.

We prove Theorem B.2 by applying Theorem A.1 to the generalized regression problem. First, we compute the leverage scores of the generalized regression problem.

**Lemma B.3.** *Let $S \subseteq [n]$ be an index. The corresponding leverage score of the generalized regression problem is*

$$\ell_S = \frac{1}{a_n}\left[|S|p_{|S|-1}^2 + (n-|S|)p_{|S|}^2 - \frac{b_n}{1+nb_n}\left(|S|p_{|S|-1} - (n-|S|)p_{|S|}\right)^2\right]. \tag{50}$$

*Proof of Lemma B.3.* Using the definition of leverage scores and Equation 47, we have

$$\ell_S = [\mathbf{A}]_S^\top (\mathbf{A}^\top \mathbf{A})^{-1}[\mathbf{A}]_S = a_n[\mathbf{A}]_S^\top \left(\mathbf{I} - \frac{b_n}{1+nb_n}\mathbf{1}\mathbf{1}^\top\right)[\mathbf{A}]_S = a_n\left(\|[\mathbf{A}]_S\|_2^2 - \frac{b_n}{1+nb_n}\left(\mathbf{1}^\top[\mathbf{A}]_S\right)^2\right). \tag{51}$$

By the definition of $\mathbf{A}$, the $\ell_2$-norm squared is given by

$$\|[\mathbf{A}]_S\|_2^2 = \frac{1}{a_n^2}\left(\sum_{i\in S}p_{|S|-1}^2 + \sum_{i\notin S}p_{|S|}^2\right) = \frac{1}{a_n^2}\left(|S|p_{|S|-1}^2 + (n-|S|)p_{|S|}^2\right) \tag{52}$$

while $\left(\mathbf{1}^\top[\mathbf{A}]_S\right)^2$ (the square of the signed sum of entries of $[\mathbf{A}]_S$ — not the square of the $\ell_1$-norm, which would instead sum absolute values) is given by

$$\left(\mathbf{1}^\top[\mathbf{A}]_S\right)^2 = \frac{1}{a_n^2}\left(\sum_{i\in S}p_{|S|-1} - \sum_{i\notin S}p_{|S|}\right)^2 = \frac{1}{a_n^2}\left(|S|p_{|S|-1} - (n-|S|)p_{|S|}\right)^2. \tag{53}$$

The lemma statement follows. $\qquad\square$

Next, we prove Theorem B.2.

*Proof of Theorem B.2.* Let $\gamma = \|\mathbf{A}\mathbf{x}^* - \mathbf{b}\|_2^2 / \|\mathbf{A}\mathbf{x}^*\|_2^2$ and suppose there are $m = O(n\log(n/\delta) + n\log(1/\delta)/\epsilon)$ samples. By the standard leverage score sampling guarantee in Theorem A.1 (see e.g., the first few lines of the proof of Corollary 3.3 for the step-by-step process), we have

$$\|\mathbf{A}\tilde{\mathbf{x}} - \mathbf{A}\mathbf{x}^*\|_2^2 \le \epsilon\|\mathbf{A}\mathbf{x}^* - \mathbf{b}\|_2^2 = \epsilon\gamma\|\mathbf{A}\mathbf{x}^*\|_2^2 \tag{54}$$

with probability $1 - \delta$. For the data matrix $\mathbf{A}$ in the unconstrained regression problem, we have

$$\|\mathbf{A}\mathbf{x}\|_2^2 = \mathbf{x}^\top \mathbf{A}^\top \mathbf{A}\mathbf{x} = \frac{1}{a_n}\mathbf{x}^\top(\mathbf{I} + b_n\mathbf{1}\mathbf{1}^\top)\mathbf{x} = \frac{1}{a_n}\left[\|\mathbf{x}\|_2^2 + b_n(\mathbf{1}^\top\mathbf{x})^2\right] \tag{55}$$

where the second equality follows by Equation 46.

Then the leverage score sampling guarantee implies

$$\frac{1}{a_n}\left[\|\mathbf{x}^* - \tilde{\mathbf{x}}\|_2^2 + b_n(\mathbf{1}^\top(\mathbf{x}^* - \tilde{\mathbf{x}}))^2\right] \le \epsilon\gamma\frac{1}{a_n}\left[\|\mathbf{x}^*\|_2^2 + b_n(\mathbf{1}^\top\mathbf{x}^*)^2\right] \tag{56}$$

so

$$\|\mathbf{x}^* - \tilde{\mathbf{x}}\|_2^2 \le \epsilon\gamma\left(\|\mathbf{x}^*\|_2^2 + b_n(\mathbf{1}^\top\mathbf{x}^*)^2\right) \le \epsilon\gamma(1 + nb_n)\|\mathbf{x}^*\|_2^2 \tag{57}$$

where the last inequality follows because $(\mathbf{1}^\top\mathbf{x}^*)^2 \le \|\mathbf{1}\|_2^2\|\mathbf{x}^*\|_2^2 = n\|\mathbf{x}^*\|_2^2$ by Cauchy-Schwarz.

Since $b_n$ is non-negative, observe that $\|\mathbf{I} + b_n \mathbf{1}\mathbf{1}^\top\|_2 = 1 + nb_n$ and $\|(\mathbf{I} + b_n \mathbf{1}\mathbf{1}^\top)^{-1}\|_2 = 1$. It follows that

$$\|\mathbf{x}^*\|_2^2 = \|(\mathbf{I} + b_n \mathbf{1}\mathbf{1}^\top)^{-1}(\mathbf{I} + b_n \mathbf{1}\mathbf{1}^\top)\mathbf{x}^*\|_2^2 = \|(\mathbf{I} + b_n \mathbf{1}\mathbf{1}^\top)^{-1}\boldsymbol{\phi}\|_2^2 \le \|(\mathbf{I} + b_n \mathbf{1}\mathbf{1}^\top)^{-1}\|_2^2 \cdot \|\boldsymbol{\phi}\|_2^2 = \|\boldsymbol{\phi}\|_2^2. \quad (58)$$

Then

$$\|\boldsymbol{\phi} - \tilde{\boldsymbol{\phi}}\|_2^2 = \|(\mathbf{I} + b_n \mathbf{1}\mathbf{1}^\top)(\mathbf{x}^* - \tilde{\mathbf{x}})\|_2^2 \tag{59}$$
$$\le \|\mathbf{I} + b_n \mathbf{1}\mathbf{1}^\top\|_2^2 \cdot \|\mathbf{x}^* - \tilde{\mathbf{x}}\|_2^2 \tag{60}$$
$$\le \epsilon\gamma(1 + nb_n)^3\|\mathbf{x}^*\|_2^2 \tag{61}$$
$$\le \epsilon\gamma(1 + nb_n)^3\|\boldsymbol{\phi}\|_2^2 \tag{62}$$

$\square$

### B.3 Preliminary Empirical Evaluation

We ran an exploratory pilot on nine weightings: Banzhaf and Beta$(\alpha, \beta)$ Shapley (Kwon & Zou, 2022a) for $(\alpha, \beta) \in \{(16, 16), (4, 4), (2, 2), (1, 1), (1, 4), (1, 16), (4, 1), (16, 1)\}$. We used the four datasets from Section 4 with $n \le 16$, where exact ground truth can be enumerated, and 10 explicands with three sampling repetitions per cell. The sample-size grid runs from $10n$ to $640n$, subject to the experiment's enumeration cap $m < 2^{n-1}$ and computational cap $m < 50{,}000$; Adult and Forest Fires therefore stop at $160n$, while Bike Sharing and Real Estate reach $640n$.

The comparison includes the unconstrained regression estimator from Lemma B.1, sampled by leverage scores with complement pairing when the weight vector is symmetric; pairing is disabled for the four asymmetric weightings, where a subset and its complement have unequal weights. Against it we run all seven baselines in our suite that accept an arbitrary weight vector $\mathbf{p}$: WSL, weighted Monte Carlo, WeightedSHAP, ARM, AME, OFA, and GELS. Within the Beta family, AME's importance sampler requires $\alpha, \beta > 1$; it is also defined for Banzhaf, where the feature-inclusion probability is fixed at $\frac{1}{2}$. AME therefore appears for Banzhaf, $(16, 16)$, $(4, 4)$, and $(2, 2)$. Figure 5 summarizes the results at $m = 40n$. Table 2 compares the observed errors with the generalized $\gamma$ and the factor $\gamma(1 + nb_n)^3$ in Theorem B.2. We compute these quantities exactly by enumerating $\mathbf{A}^{\mathrm{prob}}$ and $\mathbf{x}^*$ for all 40 explicands.

Table 2: Generalized $\gamma$, the theorem's data-dependent factor, and observed relative squared errors at $m = 40n$. The structural columns use medians over 40 explicands; the error columns additionally include three sampling repetitions per explicand. Each numeric column is summarized separately, so the fifth column (the median product) need not equal the product of the two displayed medians. "Best baseline" is selected by its median error for each weighting.

| Weighting | $nb_n$ | $\gamma$ | $(1 + nb_n)^3$ | $\gamma(1 + nb_n)^3$ | regression est. | best baseline |
|---|---|---|---|---|---|---|
| Banzhaf | 0 | 7.1 | 1.0 | 7.1 | $9.3 \times 10^{-5}$ | $2.7 \times 10^{-2}$ (WeightedSHAP) |
| Beta$(16, 16)$ | 0.17 | 7.8 | 1.6 | 13 | $2.3 \times 10^{-3}$ | $2.1 \times 10^{-2}$ (WeightedSHAP) |
| Beta$(4, 4)$ | 2.8 | 27 | 60 | $1.6 \times 10^3$ | $1.3 \times 10^{-2}$ | $1.0 \times 10^{-2}$ (WeightedSHAP) |
| Beta$(2, 2)$ | 6.7 | $1.4 \times 10^2$ | $4.8 \times 10^2$ | $6.4 \times 10^4$ | $2.6 \times 10^{-2}$ | $4.3 \times 10^{-3}$ (WeightedSHAP) |
| Beta$(1, 1)$ | 9.9 | $6.8 \times 10^2$ | $1.3 \times 10^3$ | $1.1 \times 10^6$ | $3.3 \times 10^{-2}$ | $1.5 \times 10^{-3}$ (WeightedSHAP) |
| Beta$(1, 4)$ | 10.7 | $3.3 \times 10^3$ | $1.6 \times 10^3$ | $6.7 \times 10^6$ | $1.2 \times 10^{-1}$ | $2.0 \times 10^{-2}$ (OFA) |
| Beta$(4, 1)$ | 10.7 | $3.0 \times 10^3$ | $1.6 \times 10^3$ | $6.7 \times 10^6$ | $3.3 \times 10^{-1}$ | $2.1 \times 10^{-2}$ (WeightedSHAP) |
| Beta$(16, 1)$ | 12.1 | $1.6 \times 10^4$ | $2.3 \times 10^3$ | $4.8 \times 10^7$ | $5.3 \times 10^{-3}$ | $7.4 \times 10^{-3}$ (OFA) |
| Beta$(1, 16)$ | 12.1 | $1.5 \times 10^4$ | $2.3 \times 10^3$ | $4.6 \times 10^7$ | $5.0 \times 10^{-3}$ | $8.0 \times 10^{-3}$ (OFA) |

The generalized $\gamma$ grows by more than three orders of magnitude from Banzhaf to Beta$(16, 1)$. It is a property of the weighted design, not of $v$ alone, and in this sweep it increases together with $(1 + nb_n)^3$; because the two factors move together, differences across weightings cannot be attributed to either factor alone.

The theorem does not specify its asymptotic constant, so Table 2 is a structural rather than numerical check of the guarantee. The factor $\gamma(1 + nb_n)^3$ is only weakly associated with the observed error, indicating that the bound is conservative for these instances.

At $m = 40n$, the regression estimator has $9\times$–$284\times$ lower median error for the near-uniform cases with $nb_n \leq 0.2$ and $1.4\times$–$1.6\times$ lower error for Beta$(16, 1)$ and Beta$(1, 16)$. WeightedSHAP or OFA performs better for the intermediate range $nb_n \approx 3$–$11$. Across the full sample-size grid, the near-uniform advantage persists; for the two most asymmetric weightings, the regression estimator is slightly less accurate at the smallest budgets and becomes more accurate at moderate budgets. Performance therefore depends on both the weighting and the sample budget.

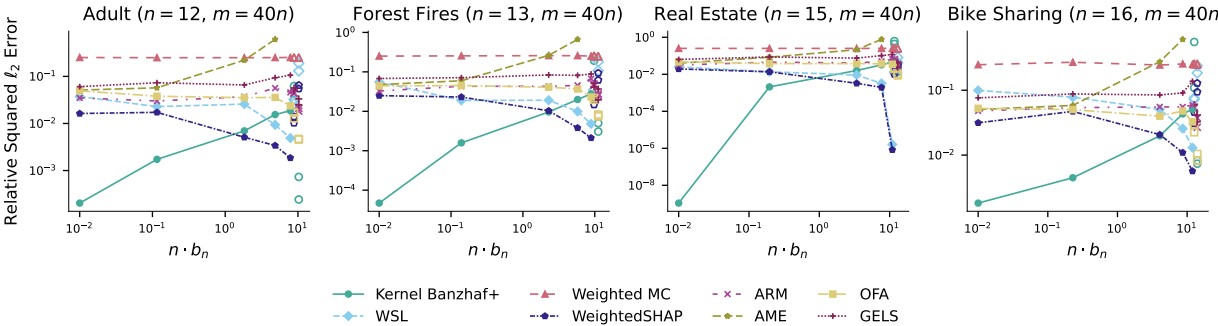

Figure 5: Exploratory pilot of the probabilistic-values extension: median relative squared error at $m = 40n$ across nine weightings on the four enumerable datasets, for the regression estimator (labeled Kernel Banzhaf+) and all seven weighting-capable baselines. Filled markers connected by lines: the symmetric Beta$(\alpha, \alpha)$ family (plus Banzhaf, plotted at $nb_n = 10^{-2}$); open markers: asymmetric weightings, which share $nb_n$ with their mirror image but differ in error. AME appears only for Banzhaf and for Beta weightings with $\alpha, \beta > 1$.

## C  Datasets and Models

**Datasets**

The Census Income dataset (Bache & Lichman, 2013; Covert & Lee, 2021), also known as the Adult dataset, involves predicting whether an individual's income exceeds \$50K/yr based on census data, using 14 features. The Forest Fires dataset (Cortez & Morais, 2007) uses meteorological data to predict burned area in Portugal's Montesinho Natural Park, with 13 features including temperature, humidity, and Fire Weather Index components. The Real Estate Valuation dataset (Yeh & Hsu, 2018) comprises market data from New Taipei City, Taiwan, with 15 features predicting house prices per unit area based on location, age, and proximity to transit. The Bike Sharing dataset (Fanaee-T & Gama, 2013) contains hourly rental counts from Capital Bikeshare in Washington D.C., with 16 features including weather conditions and temporal variables for demand prediction. The Breast Cancer Wisconsin (Diagnostic) dataset (Wolberg et al., 1993) includes 30 features computed from digitized images of fine needle aspirate, used for binary classification of breast masses as malignant or benign. The Independent Linear dataset is a synthetic regression dataset with 60 uncorrelated features, generated using the SHAP library (Lundberg & Lee, 2017). The NHANES dataset, with 79 features derived from the National Health and Nutrition Examination Survey (NHANES) I Epidemiologic Followup Study, models the risk of death over a 20-year follow-up period, as discussed in (Lundberg et al., 2020; Karczmarz et al., 2022). The Communities and Crime Unnormalized dataset (Bache & Lichman, 2013) aims to predict the total number of violent crimes per 100,000 population, comprising a predictive regression task with 101 features.

These datasets vary in size and column types and are predominantly utilized in previous studies for semi-value-based model explanation (Lundberg & Lee, 2017; Covert & Lee, 2021; Lundberg et al., 2020; Karczmarz et al., 2022). We primarily focus on tabular datasets because they are more thoroughly studied in this field and allow for easier acquisition of ground truth, especially in large datasets, using tree-based algorithms. Additionally, tabular datasets are prevalent in scenarios involving smaller datasets with fewer features.

**Models**

For the experiments involving tree set functions, we trained an XGBoost regressor model (Chen & Guestrin, 2016) with 100 trees and a maximum depth of 4. For the non-tree model experiments, we utilized a two-layer neural network equipped with a dropout layer with a rate of 0.5 to mitigate overfitting. This network was trained using a batch size of 32 and a learning rate of 0.0001, across 100 epochs. We chose this relatively simple model architecture because our primary focus is on explaining model behavior rather than maximizing its predictive accuracy.

There are generally two approaches to handling removed features in feature perturbation for general set functions, as discussed in Chen et al. (2020) and Kumar et al. (2020). Given an explicand $x$ and a subset of features $S$, define $\mathbf{x}^S$ as the observation where $\mathbf{x}_i^S = \mathbf{x}_i$ if feature $i \in S$ and, otherwise, $\mathbf{x}_i^S$ is sampled from one of two distributions. The first method involves sampling from the conditional distribution of the removed features. This approach, while precise, is computationally expensive. Alternatively, the marginal distribution can be used where the observed features $\mathbf{x}_i^S$ for $i \in S$ are ignored. Due to its lower computational complexity, we adopt the latter approach.

We compute the average of the model's predictions using replacement values randomly sampled from 50 baseline points, different from the explicand. For each explicand $\mathbf{x}$, the non-selected features in $\mathbf{x}^S$ are replaced by values from baseline points, and the average of $M(\mathbf{x}^S)$ is taken to estimate the impact of marginalizing out the non-selected features. To calculate ground truth Banzhaf values, we evaluate all $2^n$ subsets of features in this way.

## D Time Complexity

We further evaluated the computational efficiency of the Kernel Banzhaf and baseline estimators by measuring the exact time required to estimate Banzhaf values, as depicted in Figure 6. All experiments were conducted on a Lenovo SD650 with 128 GB of RAM, using only one thread for computation. While Figure 6 demonstrates that Kernel Banzhaf and MSR are fastest, we emphasize that the implementations of the estimators—especially WSL, OFA, WeightedSHAP, ARM, GELS, and AME—were not optimized for speed. MSR (paired) is omitted from Figure 6: its wall-clock times were recorded in a different compute environment than the other estimators, so they are not directly comparable. However, the paired sampling adds only a marginal overhead to the standard MSR because it only modifies how samples are drawn. A same-node microbenchmark confirms this: the runtime of MSR (paired) runtime is indistinguishable from standard MSR, as expected, since both use the same number of set-function evaluations and differ only in how the subsets are drawn.

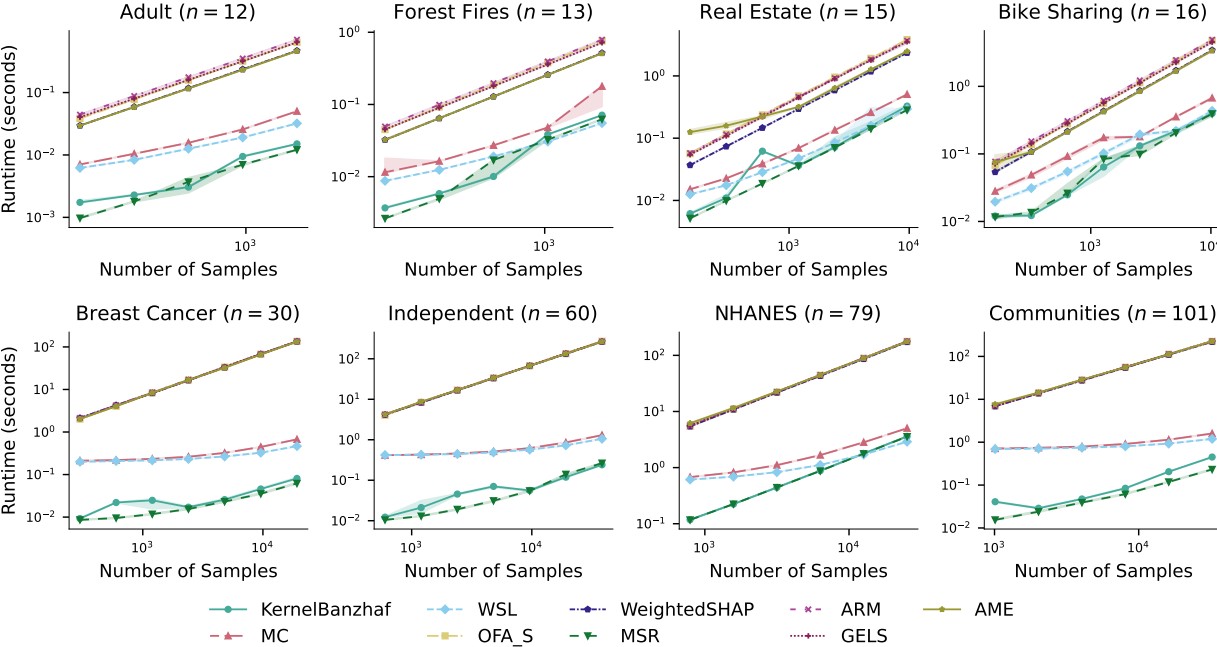

Figure 6: Computational time required for Banzhaf value estimation across varying sample sizes for eight datasets.

# E   Properties of Shapley Values and Banzhaf Values

Shapley values satisfy four desirable properties (Shapley, 1953):

- *Null Player* ensures that a player who does not contribute to any coalition, meaning their inclusion in any subset of players does not affect the overall outcome, is assigned a value of zero.

- *Symmetry* requires that two players who contribute equally to all possible coalitions receive the same value.

- *Linearity* requires that the Shapley value of a player in a combined game (formed by adding two games together) is equal to the sum of that player's Shapley values in the two individual games.

- *Efficiency* requires that the total value assigned to all players must sum to the value generated by the full set of players.

Banzhaf values three of the four properties, excluding *efficiency* which requires that the total value assigned to all players sum to the value generated by the full set of players (Banzhaf, 1965). Instead of the *Efficiency* property, the Banzhaf index satisfies *2-Efficiency*, which requires that the sum of the values of any two players equals the value of these two players when considered jointly in a reduced game setting (Banzhaf, 1965; Lehrer, 1988).

The necessity of the efficiency property has been debated in the context of machine learning. Sundararajan et al. (2017) suggest that the *Efficiency* property is only essential in contexts where semi-values, such as those in voting games, are interpreted numerically; Kwon & Zou (2022a) argues that the utility function in machine learning applications often does not correspond directly to monetary value, so aligning the sum of data values with total utility is unnecessary. In applications where the primary goal involves ranking features according to their importance or evaluating data, the exact numerical contribution of each feature is less critical. Both Banzhaf and Shapley values, despite their theoretical disparities, often yield the same ordering of players as shown in Karczmarz et al. (2022), which suffices for these applications. Therefore, given their efficiency and robustness properties, Banzhaf values serve as particularly effective tools in machine learning tasks (Wang & Jia, 2023).

## F  Feature Ranking Recovery

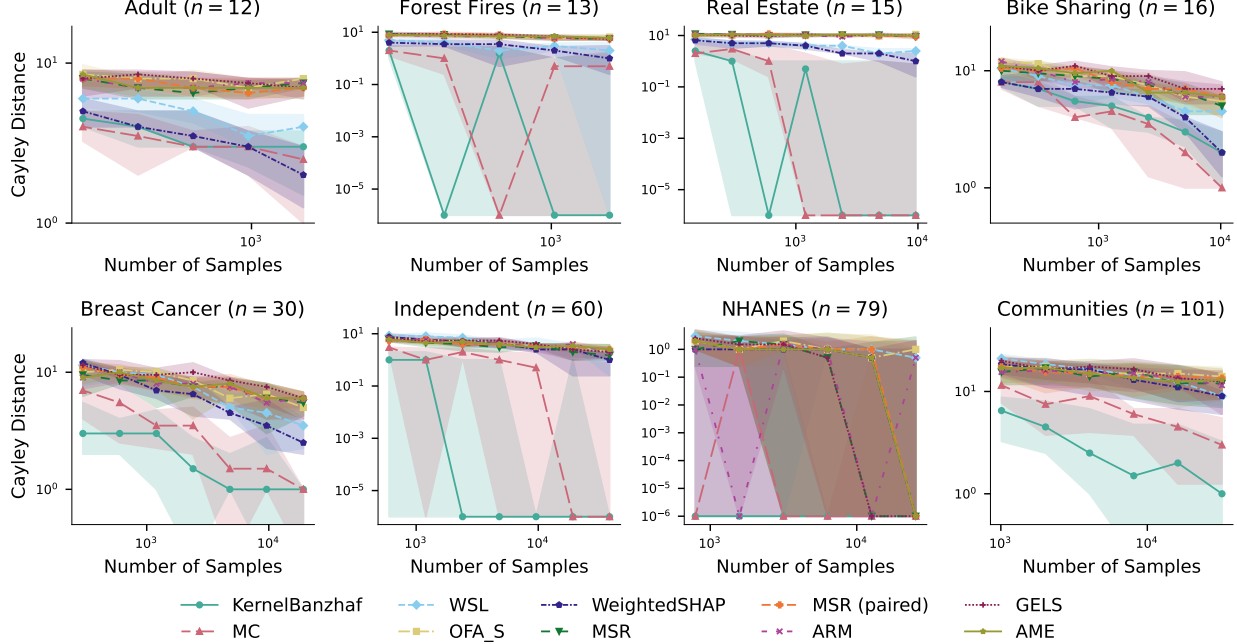

Figure 7: Comparison of non-zero Banzhaf value feature ranking recovery using Cayley distance (lower is more accurate). Kernel Banzhaf achieves the best performance in most scenarios, particularly when $n$ is large.

Aside from evaluating the quantitative errors between estimated Banzhaf values and exact Banzhaf values, another critical metric that reveals the meaningfulness of the estimated results is how well the estimator recovers feature ranking. Feature ranking is important to feature comparison and selection, which are useful for enhancing the performance of machine learning models. Accurate feature ranking helps in identifying the most influential features, thereby facilitating more efficient and effective feature engineering and dimensionality reduction strategies.

In order to evaluate this property, we incorporate two well-known metrics: *Cayley distance* and *Spearman rank correlation*. The Cayley distance refers to the minimum number of transpositions required to transform one permutation into another. This metric provides a concrete measure of the difference between two rankings, capturing the minimal edit sequence needed, which is particularly useful in understanding the stability and reliability of feature ranking methods, and it's also adopted in Karczmarz et al. (2022). Spearman's rank correlation, $\rho$, on the other hand, measures the strength and direction of association between two ranked variables. Formally, it is defined as the Pearson correlation coefficient between the rank values of the variables, mathematically expressed as:

$$\rho = 1 - \frac{6 \sum d_i^2}{n(n^2 - 1)}$$

where $d_i$ represents the difference between the ranks of corresponding variables $x_i$ and $y_i$, and $n$ is the number of observations. This metric offers insights into how well the ranking produced by the estimator preserves the monotonic relationship compared to the exact ranking, providing a measure of ranking fidelity.

In scenarios with a large feature space, the most significant features often have a more pronounced impact on model predictions. In these cases, the overall ranking may be cluttered with a large number of features that show only minor differences in their Banzhaf values, making it difficult to distinguish among lower-ranked features effectively.

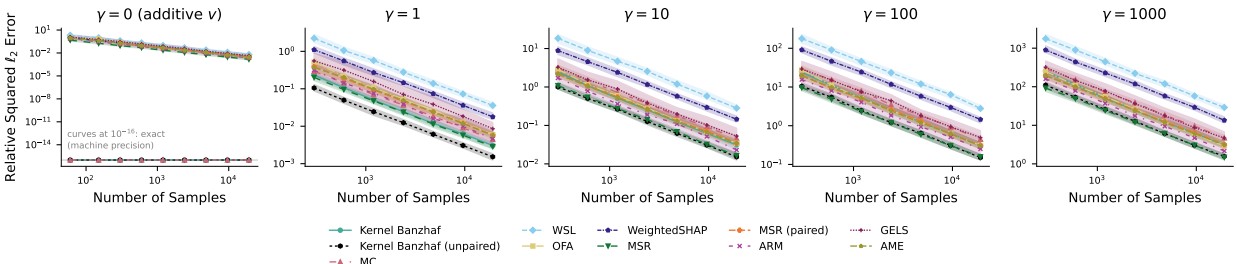

Figure 8: Synthetic set functions with purely degree-3 (odd) interactions, $n = 30$, with interaction mass varied from $\gamma = 0$ (additive) to $\gamma = 1000$ within each of three independently drawn base games. Curves show medians over the three base games $\times$ 50 sampling runs, with interquartile bands.

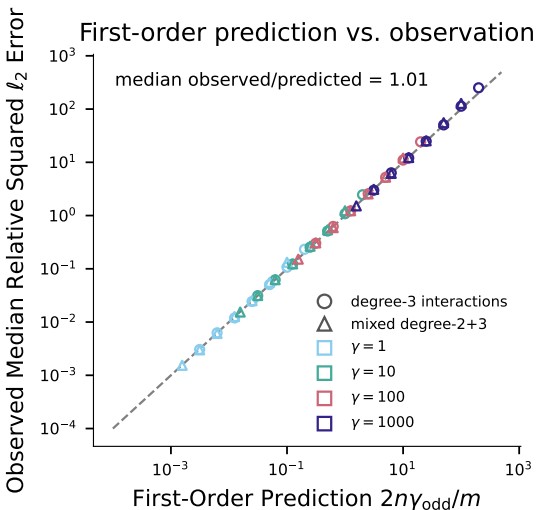

Figure 9: Observed median error of Kernel Banzhaf versus the first-order prediction $2n\gamma_{\mathrm{odd}}/m$, one point per (spectrum, $\gamma$, $m$) cell, pooling three base games. Points on the diagonal mean the prediction matches the observed error.

Focusing on features with non-zero ground truth Banzhaf values, therefore, targets those variables most likely to affect predictive accuracy and model stability, offering a more pragmatic evaluation of ranking recovery.

We present the results for this task in Figures 3 and 7. Our Kernel Banzhaf algorithm achieves the best performance across most datasets, demonstrating its effectiveness in identifying and ranking important features.

## G  Synthetic Set Functions with Controlled Interaction Mass

This appendix accompanies the discussion of Corollary 3.3's dependence on $\gamma$ in Section 3.3. We construct pseudo-Boolean Fourier games with $n = 30$, combining a fixed degree-1 component with interactions whose mass is rescaled to give $\gamma \in \{0, 1, 10, 100, 1000\}$. We set the constant Fourier coefficient to zero, so that $\gamma$ measures interaction mass rather than a constant offset, which paired sampling (or an intercept term) would remove automatically. The degree-1 coefficients determine $\phi$ in closed form, so ground-truth enumeration is unnecessary.

We consider three interaction spectra: pure degree 3, pure degree 2, and an equal mixture of degrees 2 and 3. Within each spectrum, the degree-1 coefficients, interaction sets, and coefficient directions stay fixed across

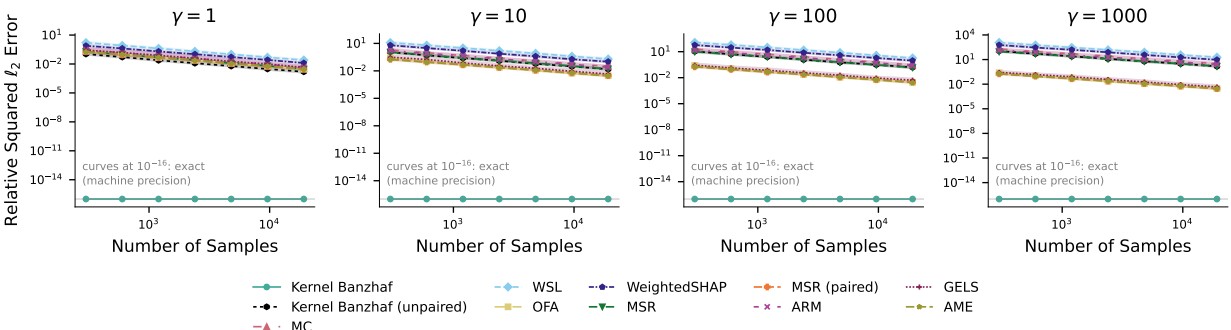

Figure 10: Synthetic set functions with purely degree-2 (even) interactions, $n = 30$. Complement pairing cancels the entire residual, so Kernel Banzhaf recovers the exact values to machine precision at every $\gamma$ (flat curves at $10^{-16}$); the unpaired variant and the other estimators follow their first-order predictions. Curves show medians over the three base games $\times$ 50 sampling runs, with interquartile bands.

the $\gamma$ sweep; only the interaction magnitude changes. We repeat the sweep for three independently drawn base games, with 50 sampling runs for each, and pool the results. The case $\gamma = 0$ is the common additive limit of all three spectra and is shown once in the degree-3 figure.

Parity matters under paired sampling. Complementing a subset flips odd-degree Fourier characters and preserves even-degree characters; this even/odd decomposition was recently used by Fumagalli et al. (2026a) to explain the efficacy of paired sampling for Shapley values, which depend only on the odd component of the set function. For Banzhaf estimation the same mechanism cuts both ways: the pair differences used by Algorithm 1 cancel the even-degree residual but double the odd-degree residual.

For a first-order calculation, let $r = \mathbf{b} - \mathbf{A}\boldsymbol{\phi}$ and let $r_{\mathrm{odd}}$ be its odd-degree part. Using $(\tilde{\mathbf{A}}^\top \tilde{\mathbf{A}})^{-1} \approx \frac{4}{m}\mathbf{I}$, $r(S) - r([n] \setminus S) = 2r_{\mathrm{odd}}(S)$, $\|\mathbf{a}_S\|_2^2 = n/4$, and $\mathbb{E}[(\mathbf{a}_S)_i r_{\mathrm{odd}}(S)] = 0$ gives

$$\frac{\mathbb{E}\,\|\hat{\boldsymbol{\phi}} - \boldsymbol{\phi}\|_2^2}{\|\boldsymbol{\phi}\|_2^2} \approx \frac{2\,n\,\gamma_{\mathrm{odd}}}{m}, \qquad \gamma_{\mathrm{odd}} := \frac{\|r_{\mathrm{odd}}\|_2^2}{\|\mathbf{A}\boldsymbol{\phi}\|_2^2}.$$

This is a first-order (in $n/m$) variance prediction for these constructions. It is consistent with, but sharper than, Corollary 3.3, whose upper bound uses the full $\gamma \geq \gamma_{\mathrm{odd}}$. Empirically, the median ratio of observed to predicted error is 1.02 (interquartile range 0.98–1.08) across all experimental cells with $\gamma_{\mathrm{odd}} > 0$ (on the degree-2 spectrum the prediction is zero and the observed error is at machine precision); Figure 9 shows the agreement after pooling the three base games.

The same calculation gives first-order relative errors $n\gamma/m$ for unpaired Kernel Banzhaf and $(n - 1 + n\gamma)/m$ for independently sampled MSR. For MSR (paired) — equivalently, the rescaling shortcut of Section 3.2 (Appendix H) — pairing doubles the degree-1 design error and the odd residual while canceling the even residual, giving $2(n - 1 + n\gamma_{\mathrm{odd}})/m$. The median observed-to-predicted ratios for these baselines range from 0.97 to 1.00. All estimators use exactly $m$ model evaluations.

Figure 8 shows the purely odd spectrum. Kernel Banzhaf's error grows approximately linearly with inter-action mass and decays as $1/m$, without a sharp transition through $\gamma = 1000$. For $\gamma \geq 100$, MSR (paired) has roughly twice the error of independently sampled MSR because it incurs the same odd-residual doubling as paired Kernel Banzhaf. On the purely even, degree-2 spectrum (Figure 10), paired sampling cancels the residual: Algorithm 1 recovers $\boldsymbol{\phi}$ to machine precision, whereas MSR follows $(n - 1 + n\gamma)/m$ and MSR (paired) remains at $2(n - 1)/m$. On the mixed spectrum (Figure 11), $\gamma_{\mathrm{odd}} = \gamma/2$, so paired and unpaired Kernel Banzhaf both have first-order error $n\gamma/m$; the MSR variants approach the regression estimators as $\gamma$ grows.

Together, the three spectra isolate the trade made by complement pairing: it cancels constant and even-degree residual components but repeats the contribution of odd-degree components. On these constructions,

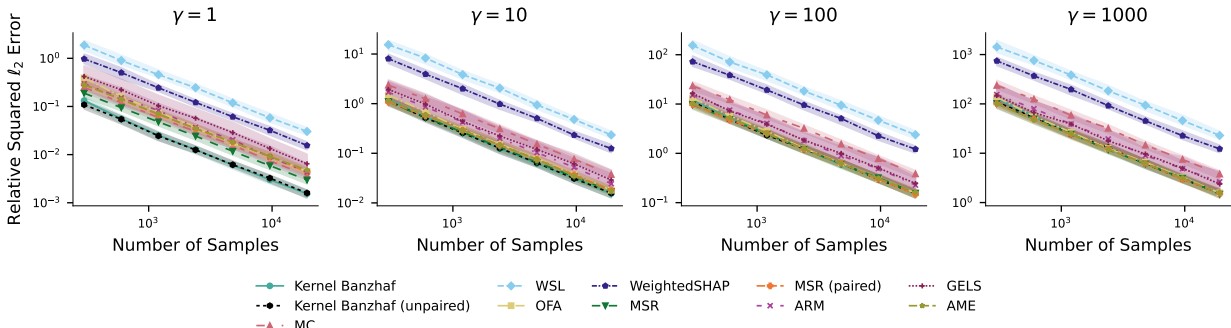

Figure 11: Synthetic set functions with interaction mass split equally between degree 2 and degree 3, $n = 30$, so that $\gamma_{\text{odd}} = \gamma/2$. The MSR variants approach the regression estimators as $\gamma$ grows. Curves show medians over the three base games $\times$ 50 sampling runs, with interquartile bands.

the first-order ratio between paired Kernel Banzhaf and MSR is

$$\frac{2n\gamma_{\text{odd}}}{n - 1 + n\gamma} < 2.$$

The ratio approaches two only when the residual is almost entirely odd and $\gamma$ is large. The degree-3 construction deliberately realizes this adverse case: for $\gamma \geq 100$, MSR has roughly half the squared error of paired Kernel Banzhaf. Higher-order effects from the sampled Gram matrix increase the observed ratio to at most 2.25 for $m \geq 40n$ and about 2.7 at $m = 10n$. The unpaired regression variant removes the pairing penalty and empirically matches MSR as $\gamma$ grows, showing that the reversal is a boundary of complement pairing rather than of the regression formulation. The enumerable real set functions lie on the favorable side of this trade: the median ratio $\gamma_{\text{odd}}/\gamma$ across the 40 explicands is $1.8 \times 10^{-4}$, and every per-dataset median is below 0.4% (Table 3).

## H   Solving the Regression Problem vs. Paired-Sampling MSR

Skipping the least-squares solve in Algorithm 1 and directly rescaling the sampled rows yields exactly the *MSR (paired)* baseline of Section 4. Here we prove this identity, compare the two estimators on identical paired samples, and analyze the additive case exactly. Let $\tilde{\mathbf{A}} \in \{\pm\frac{1}{2}\}^{m \times n}$ and $\tilde{\mathbf{b}} \in \mathbb{R}^m$ denote the paired-sampled design matrix and evaluation vector. We write the rescaled closed form as

$$\hat{\phi}_{\text{PMSR}} \;=\; \frac{4}{m}\,\tilde{\mathbf{A}}^{\top}\tilde{\mathbf{b}}.$$

The scaling $4/m$ makes the estimator unbiased: each sampled row is marginally uniform over all $2^n$ subsets, so $\mathbb{E}[\mathbf{a}_S\, v(S)] = 2^{-n}\mathbf{A}^{\top}\mathbf{b} = 2^{-n} \cdot 2^{n-2}\phi = \phi/4$ by Theorem 3.1, and hence $\mathbb{E}[\hat{\phi}_{\text{PMSR}}] = \phi$. Note that $4/m$ agrees with the population constant $1/2^{n-2}$ from Equation 5 exactly when $m = 2^n$.

### H.1   The Rescaled Closed Form Coincides with Paired-Sampling MSR

Coordinate $i$ of the rescaled closed form is

$$\left(\hat{\phi}_{\text{PMSR}}\right)_i = \frac{4}{m}\sum_{j=1}^{m}(\mathbf{a}_{S_j})_i\, v(S_j) = \frac{2}{m}\left[\sum_{j:\, i \in S_j} v(S_j) - \sum_{j:\, i \notin S_j} v(S_j)\right],$$

since $(\mathbf{a}_S)_i = +\frac{1}{2}$ if $i \in S$ and $-\frac{1}{2}$ otherwise. Under paired sampling, each of the $m/2$ pairs $(S, [n] \setminus S)$ contains feature $i$ in exactly one of its two members, so exactly $m/2$ of the sampled subsets contain $i$. The

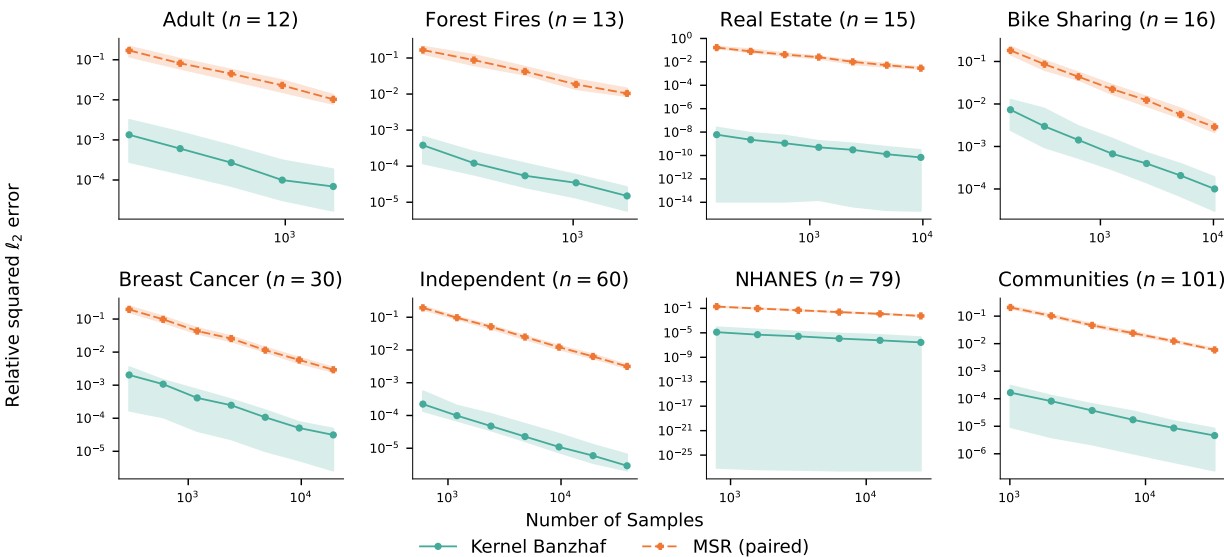

Figure 12: Matched-sample comparison of Kernel Banzhaf (Algorithm 1) and MSR (paired), i.e., the rescaled closed form $\frac{4}{m}\tilde{\mathbf{A}}^\top\tilde{\mathbf{b}}$ (Appendix H.1), computed on *identical* draws of $(\tilde{\mathbf{A}}, \tilde{\mathbf{b}})$. Points are medians of 50 runs; shaded areas are interquartile ranges.

Maximum Sample Reuse estimate computed from the same samples — the mean of $v$ over sampled subsets containing $i$ minus the mean over those not containing it — is

$$\frac{1}{m/2}\sum_{j:\,i\in S_j} v(S_j) - \frac{1}{m/2}\sum_{j:\,i\notin S_j} v(S_j),$$

which is identical, coordinate for coordinate. Hence the *MSR (paired)* baseline in Section 4 is exactly $\hat{\boldsymbol{\phi}}_{\mathrm{PMSR}}$. The identity requires pairing: with independently sampled subsets, the number of samples containing $i$ is random, and MSR normalizes by the realized counts rather than by $m/2$, so the two estimators differ. The *MSR* baseline in Section 4 is this independently sampled variant.

## H.2 Matched-Sample Comparison on Real Datasets

For each (dataset, explicand, sample size, repetition), we compute MSR (paired) and Algorithm 1 from the identical paired sample, isolating the effect of the least-squares solve. Figure 12 reports medians and interquartile ranges over 50 points per sample size (10 explicands $\times$ 5 repetitions). Algorithm 1 attains lower error on every dataset at every sample size, with median error ratios from roughly $25\times$ (Bike Sharing) to more than $10^4\times$ (NHANES). On Real Estate, the set function is numerically additive after pairing (Table 3), and least squares recovers the values to machine precision.

Table 3 reports the exact decomposition of $\gamma$ (computed by full enumeration over all $2^n$ subsets) for the four datasets small enough to enumerate. The residual Fourier mass is dominated by the constant component, which paired sampling cancels exactly; the odd-degree ($\geq 3$) share, which drives Algorithm 1's finite-sample error under paired sampling (Appendix G), is below 0.4% of the residual mass on all four datasets at the median.

## H.3 The Additive Case: Exact Recovery versus Expected Error

The starkest separation between the two estimators occurs when $v$ is exactly additive, i.e., $\mathbf{b} = \mathbf{A}\boldsymbol{\phi}$ lies in the column span of $\mathbf{A}$ ($\gamma = 0$). Then the least-squares residual is identically zero, so Algorithm 1 recovers

Table 3: Exact decomposition of $\gamma$ by full enumeration (medians over 10 explicands per dataset). $\gamma_{\mathrm{odd}}$ is the residual Fourier mass in odd degrees $\geq 3$, relative to the degree-1 mass; "const. share" is the fraction of the residual mass in the constant component.

| Dataset | $n$ | median $\gamma$ | median $\gamma_{\mathrm{odd}}$ | const. share | odd share |
|---|---|---|---|---|---|
| Adult | 12 | 2.75 | $5.3 \times 10^{-3}$ | 97.5% | 0.32% |
| Bike Sharing | 16 | 6.45 | $3.3 \times 10^{-2}$ | 97.5% | 0.36% |
| Forest Fires | 13 | 17.21 | $1.4 \times 10^{-3}$ | 99.7% | 0.007% |
| Real Estate | 15 | 18.21 | $2.6 \times 10^{-8}$ | 100.0% | $2 \times 10^{-7}\,\%$ |

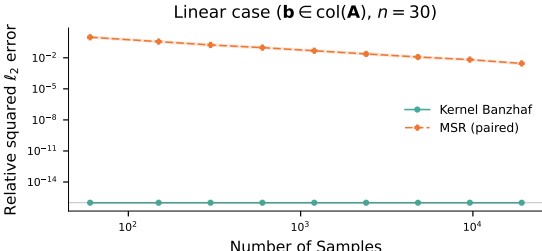

Figure 13: Matched comparison on an exactly additive game ($\mathbf{b} \in \mathrm{col}(\mathbf{A})$, $n = 30$). Kernel Banzhaf is exact once $\tilde{\mathbf{A}}$ has full column rank; MSR (paired)'s median relative squared error follows its exact expectation $2(n-1)/m$.

$\phi$ *exactly* as soon as $\tilde{\mathbf{A}}$ has full column rank — which under paired sampling happens with high probability once $m$ is a small multiple of $n$. MSR (paired) does not: its error is

$$\hat{\phi}_{\mathrm{PMSR}} - \phi = \left(\tfrac{4}{m}\tilde{\mathbf{A}}^{\top}\tilde{\mathbf{A}} - \mathbf{I}\right)\phi,$$

and this design fluctuation does not vanish at finite $m$. Its magnitude can be computed exactly. The diagonal of $\tilde{\mathbf{A}}^{\top}\tilde{\mathbf{A}}$ equals $m/4$ deterministically (every entry of $\tilde{\mathbf{A}}$ is $\pm\tfrac{1}{2}$). For $i \neq j$, under paired sampling the two rows of a pair contribute identical products, so $\tilde{\mathbf{A}}^{\top}\tilde{\mathbf{A}} = 2\sum_{p=1}^{m/2}\mathbf{a}_p\mathbf{a}_p^{\top}$ over i.i.d. pairs, giving $\mathrm{Var}\left[\tfrac{4}{m}(\tilde{\mathbf{A}}^{\top}\tilde{\mathbf{A}})_{ij}\right] = \tfrac{16}{m^2}\cdot 4\cdot\tfrac{m}{2}\cdot\tfrac{1}{16} = \tfrac{2}{m}$. Since the off-diagonal products are pairwise uncorrelated,

$$\mathbb{E}\,\|\hat{\phi}_{\mathrm{PMSR}} - \phi\|_2^2 = \sum_i\sum_{j\neq i}\tfrac{2}{m}\,\phi_j^2 = \frac{2(n-1)}{m}\,\|\phi\|_2^2,$$

i.e., an expected relative squared error of exactly $2(n-1)/m$, decaying only at the Monte Carlo rate. (Without pairing the same calculation gives $(n-1)/m$: because the degree-1 characters are odd, pairing *doubles* the design error of the rescaled closed form in the additive case.) Figure 13 confirms both predictions on a linear game with $n = 30$.

