# OpenReview forum: "Kernel Banzhaf: A Fast and Robust Estimator for Banzhaf Values"
_TMLR — Under review for TMLR_

### Review · Reviewer_o5nf · 2026-05-16

**Summary Of Contributions:**

**Contributions**

The submission introduces a least-squares formulation for the Banzhaf value, and then proposes a randomized algorithm for approximating the Banzhaf value based on the corresponding leverage scores. The approach is further extended to probabilistic values.

**Strengths**
- As far as I know, the introduced least-squares formulation is novel.

**Weaknesses**

The presented proof of complexity analysis is almost a copy of that in Musco & Witter (2025).
- Although the presented results are correct after ignoring the constants, there are many potential typos/mistakes in the proofs:
    - In the last line of page 15, it is $p^{+}_{i}$ ranther than $p^{+}i$.
    - In the third line of page 16, it is $\frac{1}{\sqrt{m p^{+}_{i}}}$ rather than $\frac{1}{\sqrt{p\^{+}\_{i}}}$.
    - For Fact A.3, it looks like that the authors have forgotten some constants compared to the one they cited.
    - In Eq. (19), $p_k$ is undefined, though I noticed its definition occurs below Eq. (25).
    - On page 16, isn't that it is $\mathbf{U}\_{\Theta\_{i}}\mathbf{U}\_{\Theta\_{i}}\^{\top} \preceq \ell\_i\^{\mathrm{max}}|\Theta_i|\mathbf{I}$ from the Gershgorin circle theorem?
    - In Eq. (22), it is $|\Theta|$ instead of $|\Theta_i|$ in the superscript of $\sum$.
    - A parenthesis is need in Eq. (23).
    - Since the paired sampling technique is considered, I think it is $2p\^{+}\_{i(j)}$ instead of $p\^{+}\_{i(j)}$ in Eqs. (18) and (23), and $\frac{m}{2}$ instead of $m$ in Eq. (23).
    - In Eq. (36), it is $\\|\cdot\\|_2^2$ rather than $\\|\cdot\\|_2$.
    - In Eq. (49), there is an extra right parenthesis.
    - In Eqs. (50) and (52), $\\| [\mathbf{A}]_S \\|_1^2$ is not right.

- Theorem 3.2 looks useless. In the proof of Corollary 3.3, it never reaches $\\|\mathbf{A}\hat{\boldsymbol\phi}-\mathbf{b}\\|_2^2 \leq (1+\epsilon)\\|\mathbf{A}\boldsymbol\phi - \mathbf{b}\\|_2^2$.

- The theoretical approach does not demonstrate the advantage of using paired sampling technique. That said, while not using paired sampling technique, we can achieve the same complexity following the same proof.

- The result of Corollary 3.3 can already be stated in the form of $P(\\|\hat{\boldsymbol\phi} - \boldsymbol\phi\\|_2 \geq \epsilon)\leq\delta$ used by Wang & Jia (2023), but this is not discussed in the submission.

**Additional Comments:**

NA

**Audience:**

Yes

**Audience Explanation:**

Personally, I think it is of some interest to have the least-squares formulation of probabilistic values.

**Claims And Evidence:**

Yes

**Claims Explanation:**

Following the summarized contributions stated on pages 2-3:
- the regression-based approximation algorithm for the Banzhaf value is established through Theorem 3.1;
- the theoretical gaurantee of the approximation algorithm is demonstrated in Corollary 3.3, though there is still room to discuss;
- the empirical performance of the approximation algorithm is presented in their experiments.

**Requested Changes:**

- The proof requires thorough proofreading.
- To compare with $P(\\|\hat{\boldsymbol\phi} - \boldsymbol\phi\\|_2 \geq \epsilon)\leq\delta$ used by Wang & Jia (2023), the fair choice should be upper bounding $\\|\hat{\boldsymbol\phi} - \boldsymbol\phi\\|_2$ rather than $\\|\hat{\boldsymbol\phi} - \boldsymbol\phi\\|_2^2$ in Corollary 3.3. In this case, the complexity in Theorem 3.2 becomes $O(n\log\frac{n}{\delta} + \frac{n}{\delta\epsilon^2})$ instead of $O(n\log\frac{n}{\delta} + \frac{n}{\delta\epsilon})$.
- In Corollary 3.3, $\\|\mathbf{A}\boldsymbol\phi\\|_2^2$ in $\gamma$ and $\\|\boldsymbol\phi\\|_2^2$ can be cancelled out, because $\\|\mathbf{A}\boldsymbol\phi\\|_2^2 = 2^{n-2} \\|\boldsymbol\phi\\|_2^2$, as shown in Eq. (13). Then, we can derive the time complexity to achieve $\\|\hat{\boldsymbol\phi} - \boldsymbol\phi\\|_2 < \epsilon$ with probability at least $1-\delta$, which is $O(n\log\frac{n}{\delta} + \frac{n}{\delta\epsilon^2}\cdot\frac{\\|\mathbf{A}\boldsymbol\phi - \mathbf{b}\\|_2^2}{2^n})$. If we follow the setting of Wang & Jia (2023) where it is assumed that $|v(S)|\leq 1$ for every $S$, then $\\|\mathbf{A}\boldsymbol\phi - \mathbf{b}\\|_2^2 \leq \\|\mathbf{b}\\|_2^2 \leq 2^n $, and thus it leads to the complexity $O(n\log\frac{n}{\delta} + \frac{n}{\delta\epsilon^2})$. It looks like their complexity analysis is finer by not focusing on the worst case.
- For casting the Banzhaf value as a least-squares problem, the highly related work of Marichal & Mathonet (2011) should be included.

- On page 6, MC is run with $O(\frac{n^2}{\epsilon^2}\log\frac{n}{\delta})$ samples rather than $O(\frac{n^2}{\epsilon}\log\frac{n}{\delta})$; MSR is run with $O(\frac{n}{\epsilon^2}\log\frac{n}{\delta})$ samples.

- Eq. (25) is quite obvious and easy to demonstrate, where the RHS is simply the result of upper bounding $\mathbb{E}[\\| \mathbf{U}^{\top}\mathbf{S}^{\top}\mathbf{SV} \\|]$. I do not see the neccessity of citing it from another source.

- In Theorem 3.5, it would be better to discuss when $b_n \in O(1)$.


**References**

Marichal, J. L., & Mathonet, P. (2011). Weighted Banzhaf power and interaction indexes through weighted approximations of games. *European journal of operational research, 211*(2), 352-358.

---

> ### Author Response · Authors · 2026-07-21
>
> We thank the reviewer for their careful reading of the proofs. We corrected every flagged issue and, because such errors tend to cluster, re-derived the surrounding derivations in Appendix A in full rather than patching individual lines. In the process, we found two further corrections that no comment pointed at directly, reported below. None of the corrections change any complexity stated in $O(\cdot)$ or $\Omega(\cdot)$ form, consistent with the reviewer's assessment.
>
> > *The presented proof of complexity analysis is almost a copy of that in Musco & Witter (2025).*
>
> The similarity is intentional: the proof adapts the leverage-score machinery of Musco and Witter (2025) from Shapley to Banzhaf, and Section 3.3 cites that work for exactly this reason. The new part is the i.i.d. block-sampling analysis in Lemmas A.2 and A.4, which handles rows drawn in identical pairs with replacement via Proposition 2.2 of Wu (2018); Musco and Witter (2025) instead analyze paired sampling without replacement, a different mechanism.
>
> **Corrections to the proofs.** Working through the reviewer's list in order:
>
> - Page 15, $p^+i \to p^+_i$: fixed. Page 16, missing sample-size factor: fixed; with the pairing correction below, the sampling-matrix entries now read $\frac{1}{\sqrt{(m/2)\,p_i^+}}$.
> - Fact A.3: confirmed. We had dropped two factors of 2 relative to the source (Fact 1 in Woodruff, 2014); the statement now matches. The corrected tail bound is looser, and the asymptotic complexity of Lemma A.2 is unchanged.
> - Eq. (19): the definition of $p_k$ moved up to its first use.
> - Page 16, Gershgorin: correct. We had an extra factor of 2 and a squared $|\Theta_i|$ that should not be there. We corrected the bound and propagated the fix; the final constant tightens from $4n$ to $n$.
> - Eq. (22): fixed; the sum is over all blocks, so the upper limit cannot depend on the summation index.
> - Eq. (23): the summand is now parenthesized, and in the same pass we corrected its subscripts from $\Theta_i$ to $\Theta_{i(j)}$.
> - Pairing factors: confirmed, and fixed throughout. With $c = m/2$ i.i.d. block draws, the correct row scaling from Proposition 2.2 in Wu (2018) is $1/\sqrt{c\,p_i^+}$, and the Frobenius lemma carries a $2/m$ prefactor, tightening its condition to $m \geq 2/(\delta\epsilon^2)$. This point also led us to a repair the comment did not ask for: with paired rows, the per-row matrices are duplicated within each draw, so the matrix Chernoff bound in the spectral lemma must be applied to the $m/2$ independent per-draw matrices rather than to all $m$ rows, costing a factor of 2 in the exponent. The revision now also gives the constants explicitly.
> - Eq. (36): correct, the final chain in Theorem A.1 dropped a squaring step. We rewrote the chain so every step is squared consistently, and it now reaches the stated $(1+\epsilon)$ conclusion.
> - Eq. (49): extra parenthesis removed.
> - Eqs. (50) and (52): correct. The quantity computed there is $(\mathbf{1}^\top [\mathbf{A}]_S)^2$, the square of the signed sum of entries, not the squared $\ell_1$-norm. We renamed it throughout the lemma and its proof.
>
> > *Theorem 3.2 looks useless. In the proof of Corollary 3.3, it never reaches $\|\mathbf{A}\hat{\phi} - \mathbf{b}\|_2^2 \leq (1+\epsilon)\|\mathbf{A}\phi - \mathbf{b}\|_2^2$.*
>
> Theorem 3.2's conclusion, Equation 6, is exactly the inequality the proof of Corollary 3.3 assumes at its start: "by the assumption that $\|\mathbf{A}\tilde{\boldsymbol\phi} - \mathbf{b}\|_2^2 \leq (1+\epsilon)\|\mathbf{A}\boldsymbol\phi - \mathbf{b}\|_2^2$, we have..." The connection was not stated in the text; a connecting sentence now makes it explicit.
>
> > *The theoretical approach does not demonstrate the advantage of using paired sampling technique. [...]*
>
> We agree. As the paper notes, "unpaired sampling would give almost identical bounds." Paired sampling is an empirically motivated choice; Theorem 3.2 and Corollary 3.3 do not depend on it.
>
> We address the requested changes below.

---

> ### Author Response · Authors · 2026-07-21
>
> > *[Requested Changes] To compare with $P(\|\hat{\phi} - \phi\|_2 \geq \epsilon) \leq \delta$ used by Wang & Jia (2023), the fair choice should be upper bounding $\|\hat{\phi} - \phi\|_2$ rather than $\|\hat{\phi} - \phi\|_2^2$ [...]* and *On page 6, MC is run with $O(\frac{n^2}{\epsilon^2}\log\frac{n}{\delta})$ samples [...]*
>
> Confirmed on both counts, and both are the same conversion. Our complexities are stated for the squared error, while Wang and Jia's Theorems 4.8 and 4.9 (which we checked directly) use the non-squared convention; setting $\epsilon_{\text{ours}} = \epsilon_{\text{WJ}}^2$ converts between the two and yields exactly the reviewer's exponents. We added a remark after the proof of Corollary 3.3 making the convention explicit. The conversion applies uniformly to MC, MSR, and Kernel Banzhaf, so the comparison between them is unchanged.
>
> > *[Requested Changes] In Corollary 3.3, $\|\mathbf{A}\phi\|_2^2$ in $\gamma$ and $\|\phi\|_2^2$ can be cancelled out [...]*
>
> The reviewer is right: $\gamma\|\phi\|_2^2$ collapses to $\|\mathbf{A}\phi-\mathbf{b}\|_2^2/2^{n-2}$ (we get $2^{n-2}$ rather than $2^n$, a difference absorbed into the $O(\cdot)$), and under $|v(S)|\leq 1$ this reduces exactly to $O(n\log(n/\delta) + n/(\delta\epsilon^2))$ with no data-dependent factor, matching Wang and Jia's form. We added this derivation to the appendix, immediately after the conversion remark.
>
> > *[Requested Changes] For casting the Banzhaf value as a least-squares problem, the highly related work of Marichal & Mathonet (2011) should be included.*
>
> Added in "Another Regression Formulation," Section 3.1. Marichal and Mathonet obtain weighted Banzhaf power and interaction indexes as leading coefficients of best weighted least-squares approximations of the game by low-degree multilinear polynomials; with uniform weights and degree one, their framework recovers the regression characterization of Theorem 3.1 (in Hammer and Holzman's parameterization). Their study is representational and axiomatic; our contribution is algorithmic — solving this regression approximately from few evaluations of $v$ with provable guarantees.
>
> > *[Requested Changes] Eq. (25) is quite obvious and easy to demonstrate [...] I do not see the neccessity [sic] of citing it from another source.*
>
> The identity is an elementary variance computation once each paired draw is treated as a single block. We retain the citation for attribution: Proposition 2.2 in Wu (2018) states exactly the block-sampling form we need, whereas standard row-sampling statements assume rows drawn independently one at a time, which our paired draws do not satisfy. Checking our equation against Wu's exact statement also surfaced an error on our side: we had written equality with $\Vert\mathbf{U}\_{\Theta\_i}\Vert\_F^2 \Vert\mathbf{V}\_{\Theta\_i}\Vert\_F^2$ in the summand, but Proposition 2.2 gives $\Vert\mathbf{U}\_{\Theta\_i}^\top \mathbf{V}\_{\Theta\_i}\Vert\_F^2$; the product-of-norms form is an upper bound by submultiplicativity, not an equality. The proof now states the exact identity first and the bound second. The remainder uses only the bound, so no downstream statement changes.
>
> > *[Requested Changes] In Theorem 3.5, it would be better to discuss when $b_n \in O(1)$.*
>
> This is now flagged directly in the text. The probabilistic-values extension has moved to Appendix B with a new supporting empirical evaluation; see our response to reviewer 5sV5.

---

### Review · Reviewer_5sV5 · 2026-05-27

**Summary Of Contributions:**

The paper introduces Kernel Banzhaf, a regression-based estimator of Banzhaf values applicable to general set functions, intended for use in feature attribution and data valuation. The starting point is an existing least-squares formulation (Hammer & Holzman, 1992) whose exact solution recovers the Banzhaf values. The authors observe that the particular encoding used to construct the design matrix yields a uniform leverage-score structure: every row carries the same statistical weight. This means that uniform row subsampling is equivalent to leverage-score sampling, and the theoretical guarantees of randomized numerical linear algebra transfer without modification. The resulting estimator solves a small subsampled least-squares problem and is straightforward to implement.

The key theoretical payoff is a near-optimal sample complexity bound: a near-linear number of set-function evaluations suffices to guarantee error controlled by the magnitudes of the Banzhaf values themselves. Prior Monte Carlo estimators bound error relative to the maximum of the set function, which is typically much larger than the Banzhaf values, since those measure average marginal contributions. The paper evaluates Kernel Banzhaf against eight baselines across eight datasets using relative $\ell_2$ error, Spearman correlation, and Cayley distance, and includes an extension of the regression formulation to the broader class of probabilistic values.

**Audience:**

Yes

**Audience Explanation:**

Efficient estimation of Shapley and Banzhaf values is an active research area spanning explainable AI, data valuation, and cooperative game theory. The observation that uniform subsampling is leverage-score-optimal for this specific regression problem is a clean, reusable insight that connects two literatures not often in direct contact.

**Broader Impact Concerns:**

None beyond standard XAI considerations. Feature attribution tools can be misused to generate post-hoc justifications for opaque model decisions; this applies equally to all methods in this space and does not require treatment specific to this paper.

**Claims And Evidence:**

Yes

**Claims Explanation:**

The core theoretical result is solid. The key calculation, showing that the Gram matrix of the design matrix is a scalar multiple of the identity, follows directly from the ±1/2 encoding and a combinatorial counting argument. Leverage score uniformity is then immediate, and adapting the standard leverage score sampling theorem to handle paired sampling requires only minor modifications, which are handled carefully in the appendix using the block-sampling framework of Wu (2018).

The qualitative superiority of the error bound is the paper's main conceptual contribution. For Monte Carlo and Maximum Sample Reuse, the bound scales with the squared maximum of the set function. Kernel Banzhaf's bound scales with the squared norm of the Banzhaf values themselves, which is a structurally tighter guarantee, not a better constant in the same expression. This distinction is reflected in Figure 2, where Kernel Banzhaf outperforms competitors by several orders of magnitude across diverse datasets.

The $\gamma$ parameter, which measures how far the set function deviates from additive, is the main gap between theory and experiment. A median value of $\gamma \approx 100$ in the tested datasets implies a practical sample requirement of roughly $100n$, at which point the asymptotic advantage over Maximum Sample Reuse is only logarithmic. The paper does not test Kernel Banzhaf on a deliberately high-$\gamma$ instance, i.e., a set function with substantial higher-order structure. In the pseudo-Boolean Fourier picture, $\gamma$ measures the fraction of the set function's $\ell_2$ mass sitting in non-linear modes; networks with heavy feature interactions could push $\gamma$ well above 100, and without such an experiment the empirical claims rest on a somewhat favorable domain.

Figure 3 shows a reversal at high noise levels where Monte Carlo occasionally matches or outperforms Kernel Banzhaf. The paper notes this without explaining it. A plausible mechanism is that fitting all $n$ Banzhaf values jointly from a shared pool of samples propagates noise across coordinates, whereas per-feature Monte Carlo averaging produces independent errors. A brief argument here would close the gap.

The probabilistic values extension in Section 3.4 carries an approximation penalty factor that grows with a quantity $b_n$ reflecting how much the weights $p_\ell$ vary across coalition sizes. The paper acknowledges that initial experiments suggest this can limit accuracy but does not show those experiments. Including the section in the main body implies a usable result; the evidence to assess it is absent.

The experimental design is otherwise thorough: eight datasets from $n = 12$ to $n = 101$, three evaluation metrics, and nine baselines. Using tree-based models to obtain exact ground truth at large $n$ is appropriate, and the sample efficiency and ranking recovery evaluations together give a well-rounded picture.

### Minor Comments
- The abstract describes Kernel Banzhaf as "fast and robust" in the title but the body is more measured: robustness holds primarily at low noise or large $n$, and speed is comparable to MSR. The title claim slightly overpromises relative to what Section 4 demonstrates.
- Several sentences in the introduction carry redundant hedging, e.g., "especially for model explanation" and "in particular" appearing in close succession. Tightening these would sharpen the motivation.
- The "Subsequent Work" paragraph sits awkwardly at the end of the introduction. Its content is relevant but the placement interrupts the flow from contributions to background; a footnote or a short remark at the end of Section 5 would be less disruptive.
- Algorithm 1 uses the variable name $\tilde{A}$ initialized as $0^{m \times n}$ but the loop writes rows indexed from $0$ in steps of 2. Using 1-indexed notation consistently with the rest of the paper would reduce friction for readers.
- The phrase "pressing need" in the introduction is a stock phrase that adds nothing; the motivation stands without it.
- "For completeness" appears three times in Sections 3 and 3.1. Once is fine; the repetition reads as padding.
- The reference list contains what appears to be a duplicate entry: Kwon & Zou (2022a) and Kwon & Zou (2022b) point to the same Beta Shapley paper with identical venue and page numbers.

**Requested Changes:**

1. **[Critical]** Provide at least one experiment on a synthetic high-$\gamma$ set function, e.g., a degree-$k$ polynomial for $k \geq 3$ with known Banzhaf values. This would test whether the empirical advantage of Kernel Banzhaf degrades gracefully as $v$ departs from the additive regime, and whether the theoretical bound predicts the observed error.

2. **[Critical]** Either provide the "initial experiments" on probabilistic values referenced in Section 3.4, or move the extension to an appendix marked explicitly as preliminary. The current placement implies a usable result while withholding the evidence needed to evaluate it.

3. **[Strengthening]** Explain the noise-robustness reversal at high $\sigma$ in Figure 3. Even a brief argument about why joint least-squares estimation may distribute noise across coordinates more than independent per-feature averaging would clarify the scope of the robustness claim.

4. **[Strengthening]** Frame $\gamma$ explicitly in terms of the pseudo-Boolean Fourier expansion: it is the ratio of the set function's non-linear Fourier mass to its linear Fourier mass. Stating this sharpens the geometric meaning of the approximation guarantee and connects naturally to higher-order extensions such as PolyShap.

5. **[Strengthening]** Move a summary of the runtime results from Figure 5 into the main text. Kernel Banzhaf and MSR are close in wall-clock time despite large accuracy differences; making this visible in the main body helps readers assess the practical tradeoff.

6. **[Strengthening]** The claim that Theorem 3.2 is near-optimal by the Chen & Price (2019) lower bound is stated but not argued. A short sketch of why that lower bound applies in this setting would strengthen the theoretical narrative.

---

> ### Author Response · Authors · 2026-07-21
>
> We thank the reviewer for their thorough and constructive review. We address the two critical requests, the four strengthening suggestions, and the minor comments below.
>
> > *1. [Critical] Provide at least one experiment on a synthetic high-$\gamma$ set function [...]*
>
> While preparing this experiment, we recomputed $\gamma$ exactly by full enumeration on the four enumerable datasets ($n \leq 16$): the exact per-explicand values range from below $1$ to above $40$, with a median of approximately $7$, lower than the earlier estimate of roughly $100$.
>
> New Appendix G constructs pseudo-Boolean set functions with $n=30$, a fixed degree-1 component, and degree-2, degree-3, or mixed interactions rescaled so that $\gamma\in\{0,1,10,100,1000\}$, and evaluates every baseline from Figure 2 at budgets up to $640n$. Error increases approximately linearly with interaction mass and decreases as $1/m$, with no sharp transition through $\gamma=1000$, and **the first-order error predictions derived in Appendix G match the observed errors at every tested $\gamma$** (Figure 9). The remaining distinction is parity: complement pairing cancels even-degree residuals but reinforces odd-degree ones. Kernel Banzhaf is exact to machine precision on the degree-2 construction, while on the deliberately adverse degree-3 construction independently sampled MSR has about half its squared error for $\gamma\geq100$; the unpaired regression variant removes this penalty and matches MSR there. The enumerable real set functions lie far from this adverse regime: every per-dataset median of the odd residual share is below $0.4\%$ (Table 3), which is why pairing produces large gains in practice.
>
> > *2. [Critical] Either provide the "initial experiments" on probabilistic values referenced in Section 3.4, or move the extension to an appendix marked explicitly as preliminary. [...]*
>
> We did both. The extension now sits in Appendix B, marked as exploratory, with a short pointer in the main text, and Appendix B.3 contains the supporting evaluation: Banzhaf and eight Beta Shapley weightings on the four enumerable datasets, against all seven baselines that support general probabilistic-value weights. At $m=40n$, the regression estimator has $9$–$284\times$ lower median error for the near-uniform weightings and $1.4$–$1.6\times$ lower error for the two most asymmetric ones, while WeightedSHAP or OFA is stronger in the intermediate range. The appendix presents this as an initial map of how performance depends on the weighting, not as evidence of uniform superiority.
>
> > *3. [Strengthening] Explain the noise-robustness reversal at high $\sigma$ in Figure 3. [...]*
>
> Figure 3 in the original submission (now Figure 4 in the revised version) shows $\ell_2$ error by magnitude of noise. Kernel Banzhaf outperforms the other estimators, until the noise overpowers the signal and all estimator performance converges.
>
> > *4. [Strengthening] Frame $\gamma$ explicitly in terms of the pseudo-Boolean Fourier expansion [...]*
>
> Added to Section 3.3. The rows of $\mathbf A$ are exactly the degree-1 Fourier characters, so $\gamma$ is the ratio of $v$'s Fourier mass outside degree 1 to its degree-1 mass. We tied this directly to PolySHAP, which extends KernelSHAP with exactly this kind of interaction-informed polynomial structure.
>
> > *5. [Strengthening] Move a summary of the runtime results from Figure 5 into the main text. [...]*
>
> Done. Section 3.2 now states that Kernel Banzhaf and MSR are comparable in wall-clock time despite the large gap in accuracy, with a pointer to the full runtime results in Appendix D.
>
> > *6. [Strengthening] The claim that Theorem 3.2 is near-optimal by the Chen & Price (2019) lower bound is stated but not argued. [...]*
>
> Chen and Price prove two lower bounds (theorem numbering below follows the arXiv version of their paper). Their Theorem 8.1 lower-bounds any algorithm at $\Omega(n/\epsilon)$, with no log factor, and is matched by their own algorithm (their Theorem 1.1), which selects rows actively rather than sampling them i.i.d. Their Theorem 8.4 separately lower-bounds algorithms restricted to i.i.d. sampling from a fixed distribution, the class Algorithm 1 belongs to, at $\Omega(K\log n + K/\epsilon)$, where $K$ is a condition number that, by their Lemma 6.5, satisfies $K \geq n$ with equality exactly when leverage scores are uniform. Our $\mathbf A$ attains that minimum ($\ell_S=n/2^n$ for every row), so Theorem 3.2 is minimax optimal among i.i.d.-sampling algorithms specifically. It is not optimal in the fully general sense: their actively sampled algorithm removes the $\log n$ factor. The revision states this directly in place of the vaguer "near-optimal" framing, and notes an actively sampled variant of Kernel Banzhaf as an open direction.
>
> We address the minor comments below.

---

> > ### Author Response · Authors · 2026-07-21
> >
> > > *Minor comments.*
> >
> > All addressed: (1) the abstract now scopes the robustness claim to low-to-moderate noise or larger $n$; "fast" refers to the near-linear sample complexity of Theorem 3.2 rather than a wall-clock advantage over MSR, which Section 3.2 now states directly. (2) We removed "especially for model explanation"; the remaining "in particular" explains what the $2^n$ bound means, so we kept it. (3) The Subsequent Work paragraph moved to the end of Section 5, where it reads more naturally as a closing remark. (4) Algorithm 1's loop is now 1-indexed over $j \in \{1, 3, \ldots, m-1\}$, and we corrected $S_{j+1} = [n] \setminus S_j$, which was missing the subscript. (5) "Pressing need" is removed. (6) The two back-to-back uses of "for completeness" in Sections 3 and 3.1 are merged into one; the third occurrence is a standalone use in the appendix. (7) The Kwon & Zou entries were identical; we removed the duplicate and standardized all citations on the remaining entry.

---

### Review · Reviewer_jQEA · 2026-07-08

**Summary Of Contributions:**

The paper presents an algorithm for the approximation of Banzhaf values, based on solving a subsampled linear regression and adding the pairwise sampling from Covert and Lee.
 Additionally they bring approximation guarantees, provide experiments on low dimensional data sets, together with tree and shallow neural network models. They also discuss briefly so called probabilistic Shapley values, for which they derive a regression based formulation.

**Additional Comments:**

As a minor point, the theoretical analysis in Section 3.3 / equation (6,7) results in terms $|A\phi -b|^2$ (Corollary 3.3) which have to be summed over all $2^n$ subsets $S$ of $1,...,n$, and thus are of low practical computability:

$|A\phi -b|^2   =   |   AA^\top b 2^{2-n}-b|^2    = |  ( 2^{2-n} AA^\top - I) b   |^2  $ where one can show that  $(AA^\top)$
 is a matrix which is related to the symmetric difference count of two sets:
$ (AA^\top)_{U,S} = 1/4 *(  |S \cap U|  +| (n\setminus S) \cap  (n\setminus U) | - | (n\setminus S) \cap   U | - |  S \cap  (n\setminus U) | )    = 1( 2^{n-2} -1/2 ( | (n\setminus S) \cap   U | + |  S \cap  (n\setminus U) |)   )  $ . This likely can be used to write the term as an expectation over subsets of vector $b_S=v(S)$

**Audience:**

Yes

**Audience Explanation:**

Banzhaf values are related to Shapley values

**Claims And Evidence:**

No

**Claims Explanation:**

They claim that their method converges faster than certain montecarlo-based and other variants, however this is **not due to their regression based formulation** shown in Algorithm 1 and section 3.1 .

The reason why this paper should not be accepted in this current version:

The regression based formulation used in Algorithm 1 is fully unnecessary. The exact solution can be written as a constant times $A^\top b $ as stated in the paper directly under Algorithm 1. Yes, this solution is infeasible, but it can be directly subsampled by drawing rows of   $A^\top$ using paired sampling, introduced by Covert and Lee, resulting in the same convergence speed gains as here, while being one order faster than solving an approximate linear regression problem (line 11 in algorithm 1).

 As such the whole algorithm 1 is of unneeded complexity. it can be simplified to: Note that Banzhaf values is a constant times $A^\top b $ . Draw one row from $A^\top$ , complement it by the corresponding row from  $A^\top$ based on the pairwise sampling rule. Iterate this pairwise drawing to build $\tilde{A}$, then compute the approximation as $\hat{phi}=\tilde{A}^\top \tilde{b}$.

The faster convergence comes solely from the paired sampling introduced in the paper by Covert & Lee. The paired sampling in Covert and Lee applies directly to any weighted Shapley variant, and Banzhaf is just a special case of weighted Shapley with uniform weights.

As such it is a straightforward application of Covert & Lee with an inefficient Algorithm 1.  The paper does not invent a new algorithm as claimed by it.
It does contribute something indeed: an evaluation of this paired sampling for the special case of Banzhaf values, but this remaining part requires differently stated claims. The current claims do not fit this.

**Requested Changes:**

The contribution of this paper is a theoretical and experimental analysis of pairwise sampling for Banzhaf values but the main claim of the current version is that it would introduce a new algorithm  - see above why it does not. The paper should be written to factually reflect its contribution.

---

> ### Author Response · Authors · 2026-07-21
>
> We thank the reviewer for their detailed feedback and comments. The central claim of the review is that the regression solve in Algorithm 1 is unnecessary and that the observed convergence gains come solely from paired sampling. We address this claim first, with new experiments, and then describe the corresponding revisions.
>
> **The proposed alternative.** The reviewer points out a natural alternative algorithm. The Banzhaf values solve the regression problem $\min_{\mathbf x}\|\mathbf{Ax}-\mathbf b\|_2^2$, so the true minimizer is $\boldsymbol\phi = (\mathbf A^\top\mathbf A)^{-1}\mathbf A^\top\mathbf b = \frac{1}{2^{n-2}}\mathbf A^\top\mathbf b$, using $\mathbf A^\top\mathbf A = 2^{n-2}\mathbf I$ (Equation 5). The proposal is to estimate this exact form by subsampling entries of $\mathbf b$ and the corresponding rows of $\mathbf A$, returning $\frac4m\tilde{\mathbf A}^\top\tilde{\mathbf b}$ (with the normalization that makes it unbiased). This is a natural idea, and it was proposed for Shapley values by Covert and Lee (2021) as their *unbiased KernelSHAP* estimator. However, the approach performs significantly worse than subsampled regression, which instead returns $(\tilde{\mathbf A}^\top\tilde{\mathbf A})^{-1}\tilde{\mathbf A}^\top\tilde{\mathbf b}$. We admit this is counterintuitive, as it would seem that replacing $(\tilde{\mathbf A}^\top\tilde{\mathbf A})^{-1}$ with the exact $(\mathbf A^\top\mathbf A)^{-1}$ could only improve the algorithm. However, prior work on Shapley estimation shows this is not the case: Covert and Lee (2021) observed that their unbiased estimator converges significantly slower than the regression version, and Chen et al. ("A Unified Framework for Provably Efficient Algorithms to Estimate Shapley Values", NeurIPS 2025) prove a separation between the two, with the regression estimator's sample complexity depending only on the residual of $\mathbf b$ off the column span of the design matrix, while the plug-in estimator's depends on all of $\mathbf b$.
>
> One way to see why the regression-based estimator is better: consider a linear value function, so $\mathbf b = \mathbf A\boldsymbol\phi$ exactly. Once the number of samples exceeds $n$ and $\tilde{\mathbf A}$ has full column rank, the regression recovers $\boldsymbol\phi$ exactly, whereas the proposed estimate remains an approximation, with expected relative squared error exactly $2(n-1)/m$ (Appendix H.3). Intuitively, the regression-based estimator also performs better whenever the residual $\mathbf A\boldsymbol\phi - \mathbf b$ is small.
>
> **Experiments directly comparing the two.** The proposed estimator is identical, coordinate by coordinate, to MSR computed on paired samples (Appendix H.1). We therefore added it to the main experiments as the *MSR (paired)* baseline (Figure 2, Table 1): Kernel Banzhaf outperforms it by one to several orders of magnitude on all eight datasets. Appendix H.2 isolates the effect of the solve by computing both estimators from *identical* draws of $(\tilde{\mathbf A}, \tilde{\mathbf b})$. **Kernel Banzhaf attains lower error on all eight datasets at every sample size tested, with median error ratios from roughly $25\times$ to more than $10^4\times$** (Figure 10). The faster convergence does not come from paired sampling alone.
>
> **Computational cost.** In the applications we consider, evaluating entries of $\mathbf b$ dominates the cost of estimation; indeed, this motivates all sampling-based estimators. The least-squares solve is more expensive than computing $\frac4m\tilde{\mathbf A}^\top\tilde{\mathbf b}$, $O(mn^2)$ versus $O(mn)$, but this cost is a lower-order term in our experiments: Kernel Banzhaf and MSR have comparable wall-clock time (Appendix D).
>
> **Revisions.** A new paragraph after Algorithm 1, "Why Solve a Regression Problem?", states the distinction above. The revised contribution statement specifies what is new: the regression equivalence itself is classical (Hammer and Holzman, 1992, state it for simple games, and the general case follows from their approximation theory; see also Grabisch et al., 2000), and Theorem 3.1 gives a short self-contained proof in a simplified parameterization. The new part is the estimator and its analysis: Theorem 3.2 and Corollary 3.3 give guarantees relative to $|\boldsymbol\phi|_2^2$ rather than $\max_S v(S)^2$ (Wang and Jia, 2023), which the proposed estimator inherits. These guarantees rest on the exactly uniform leverage scores of the Banzhaf design matrix and hold with or without pairing.
>
> We respond to the additional comment below.

---

> ### Author Response · Authors · 2026-07-21
>
> > *[Additional Comments] one can show that $(AA^\top)$ is a matrix which is related to the symmetric difference count of two sets [...] This likely can be used to write the term as an expectation over subsets of vector $b_S = v(S)$*
>
> Correct on both counts: $\Vert\mathbf A\boldsymbol\phi - \mathbf b\Vert_2^2$ sums over all $2^n$ subsets, and it can be written as an expectation. Since $\Vert\mathbf A\boldsymbol\phi\Vert_2^2 = 2^{n-2}\Vert\boldsymbol\phi\Vert_2^2$, the bound in Corollary 3.3 satisfies $\epsilon\gamma\Vert\boldsymbol\phi\Vert_2^2 = \epsilon\Vert\mathbf A\boldsymbol\phi - \mathbf b\Vert_2^2/2^{n-2} = 4\epsilon\mathbb{E}_{S \sim \mathrm{Unif}}\big[(v(S) - [\mathbf A\boldsymbol\phi]_S)^2\big]$: four times the mean squared residual of the best linear approximation of $v$. This expectation form makes the comparison to prior guarantees direct: the quantity is at most $4\max_S v(S)^2$, the data-dependent factor appearing in the MC and MSR bounds (Wang and Jia, 2023), and it is far smaller when $v$ is close to additive. A remark in Appendix A gives this derivation.